# LiteTransformerSearch: Training-free Neural Architecture Search for Efficient Language Models

**Mojan Javaheripi**[1], **Gustavo H. de Rosa**[2], **Subhabrata Mukherjee**[2],
**Shital Shah**[2], **Tomasz L. Religa**[3], **Caio C.T. Mendes**[2],
**Sebastien Bubeck**[2], **Farinaz Koushanfar**[1], **Debadeepta Dey**[2]

[1]University of California San Diego, [2]Microsoft Research, [3]Microsoft

`mojan@ucsd.edu, dedey@microsoft.com`

## Abstract

The Transformer architecture is ubiquitously used as the building block of large-scale autoregressive language models. However, finding architectures with the optimal trade-off between task performance (perplexity) and hardware constraints like peak memory utilization and latency is non-trivial. This is exacerbated by the proliferation of various hardware. We leverage the somewhat surprising empirical observation that the number of decoder parameters in autoregressive Transformers has a high rank correlation with task performance, irrespective of the architecture topology. This observation organically induces a simple Neural Architecture Search (NAS) algorithm that uses decoder parameters as a proxy for perplexity without need for any model training. The search phase of our training-free algorithm, dubbed Lightweight Transformer Search (LTS)[1], can be run directly on target devices since it does not require GPUs. Using on-target-device measurements, LTS extracts the Pareto-frontier of perplexity versus any hardware performance cost. We evaluate LTS on diverse devices from ARM CPUs to NVIDIA GPUs and two popular autoregressive Transformer backbones: GPT-2 and Transformer-XL. Results show that the perplexity of 16-layer GPT-2 and Transformer-XL can be achieved with up to $1.5\times, 2.5\times$ faster runtime and $1.2\times, 2.0\times$ lower peak memory utilization. When evaluated in zero and one-shot settings, LTS Pareto-frontier models achieve higher average accuracy compared to the 350M parameter OPT across 14 tasks, with up to $1.6\times$ lower latency. LTS extracts the Pareto-frontier in under 3 hours while running on a commodity laptop. We effectively remove the carbon footprint of hundreds of GPU hours of training during search, offering a strong simple baseline for future NAS methods in autoregressive language modeling.

## 1 Introduction

The Transformer architecture [42] has been used as the de-facto building block of most pre-trained language models like GPT [5]. A common problem arises when one tries to create smaller versions of Transformer models for edge or real-time applications (e.g. text prediction) with strict memory and latency constraints: it is not clear what the architectural hyperparameters should be, e.g., number of attention heads, number of layers, embedding dimension, and the inner dimension of the feed forward network, etc. This problem is exacerbated if each Transformer layer is allowed the freedom to have different values for these settings. This results in a combinatorial explosion of architectural hyperparameter choices and a large heterogeneous search space. For instance, the search space considered in this paper consists of over $10^{54}$ possible architectures.

Neural Architecture Search (NAS) is an organic solution due to its ability to automatically search through candidate models with multiple conflicting objectives like latency vs. task performance. The central challenge in NAS is the prohibitively expensive function evaluation, i.e., evaluating each architecture requires training it on the dataset at hand. Thus it is often infeasible to evaluate more

---

[1]code available at `https://github.com/microsoft/archai/tree/neurips_lts/archai/nlp`

36th Conference on Neural Information Processing Systems (NeurIPS 2022).

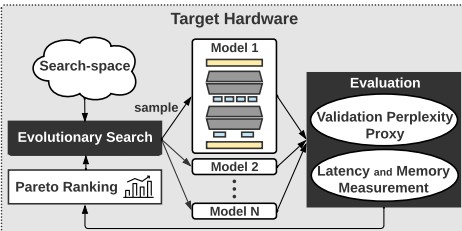

Figure 1: High-level overview of LTS. We propose a training-free zero-cost proxy for evaluating the validation perplexity of candidate architectures. Pareto-frontier search is powered by evolutionary algorithms which use the proposed proxy along with real latency and memory measurements on the target hardware to evaluate sampled architectures.

than a handful of architectures during the search phase. Supernets [31] have emerged as a dominant paradigm in NAS which combine all possible architectures into a single graph and jointly train them using weight-sharing. Nevertheless, supernet training imposes constraints on the expressiveness of the search space [29] and is often memory-hungry [52, 6, 51] as it creates large networks during search. Additionally, training supernets is non-trivial as children architectures may interfere with each other and the ranking between sub-architectures based on task performance is not preserved [29][2].

We consider a different approach by proposing a training-free proxy that provides a highly accurate ranking of candidate architectures during NAS without need for costly function evaluation or supernets. Our scope is NAS for efficient autoregressive Transformers used in language modeling. We design a lightweight search method that is target hardware-aware and outputs a gallery of models on the Pareto-frontier of perplexity versus hardware metrics. We term this method Lightweight Transformer Search (LTS). LTS relies on our somewhat surprising observation: *the decoder parameter count has a high rank correlation with the perplexity of fully trained autoregressive Transformers.*

Given a set of autoregressive Transformers, one can accurately rank them using decoder parameter count as the proxy for perplexity. Our observations are also well-aligned with the power laws in [22], shown for homogeneous autoregressive Transformers, i.e., when all decoder layers have the same configuration. We provide extensive experiments that establish a high rank correlation between perplexity and decoder parameter count for *both* homogeneous and heterogeneous search spaces.

The above phenomenon coupled with the fact that a candidate architecture's hardware performance can be measured on the target device leads to a training-free search procedure: *pick one's favorite discrete search algorithm (e.g. evolutionary search), sample candidate architectures from the search space; count their decoder parameters as a proxy for task performance (i.e., perplexity); measure their hardware performance (e.g., latency and memory) directly on the target device; and progressively create a Pareto-frontier estimate.* While we have chosen a reasonable search algorithm in this work, one can plug and play any Pareto-frontier search method such as those in [20].

Building upon these insights, Figure 1 shows a high-level overview of LTS. We design the first training-free Transformer search that is performed entirely on the target (constrained) platform. As such, LTS easily performs a multi-objective NAS where several underlying hardware performance metrics, e.g., latency and peak memory utilization, are simultaneously optimized. Using our training-free proxy, we extract the 3-dimensional Pareto-frontier of perplexity versus latency and memory in a record-breaking time of $< 3$ hours on a commodity Intel Core i7 CPU. Notably, LTS eliminates the carbon footprint from hundreds of GPU hours of training associated with legacy NAS methods.

To corroborate the effectiveness of our proxy, we train over 2900 Transformers on three large language modeling benchmark datasets, namely, WikiText-103 [27], One Billion Word [7], and Pile [17]. We use LTS to search for Pareto-optimal architectural hyperparameters in two popularly used autoregressive Transformer backbones, namely, Transformer-XL [10] and GPT-2 [32]. We believe decoder parameter count should be regarded as a competitive baseline for evaluating Transformer NAS, both in terms of ranking capabilities and easy computation. We open-source our code along with tabular information of our trained models to foster future NAS research on Transformers.

## 2   Related Work

Here, we discuss literature on automated search for Transformer architectures in the language domain. We refer to extensive surveys on NAS [14, 49] for a broader overview of the field.

**Decoder-only Architectures.** So et al. [37] search over TensorFlow programs that implement an autoregressive language model via evolutionary search. Since most random sequences of programs

---

[2] See [29] for a comprehensive treatment of the difficulties of training supernets.

either have errors or underperform, the search has to be seeded with the regular Transformer architecture, termed "Primer". As opposed to "Primer" which uses large computation to search a general space, we aim to efficiently search the "backbone" of traditional decoder-only Transformers. Additionally, the objective in "Primer" is to find models that train faster. Our objective for NAS, however, is to deliver Pareto-frontiers for inference, with respect to perplexity and hardware constraints.

**Encoder-only Architectures.** Relative to decoder-only autoregressive language models, encoder-only architectures like BERT [11] have received much more recent attention from the NAS community. NAS-BERT [50] trains a supernet to efficiently search for masked language models (MLMs) which are compressed versions of the standard BERT, Such models can then be used in downstream tasks as is standard practice. Similar to NAS-BERT, Xu et al. [51] train a supernet to conduct architecture search with the aim of finding more efficient BERT variants. They find interesting empirical insights into supernet training issues like differing gradients at the same node from different child architectures and different tensors as input and output at every node in the supernet. The authors propose fixes that significantly improve supernet training. Tsai et al. [41], Yin et al. [53], Gao et al. [16] also conduct variants of supernet training with the aim of finding more efficient BERT models.

**Encoder-Decoder Related:** Applying the well-known DARTS [24] approach to Transformer search spaces leads to memory-hungry supernets. To mitigate this issue, Zhao et al. [61] propose a multi-split reversible network and a memory-efficient backpropagation algorithm. One of the earliest papers that applied discrete NAS to Transformer search spaces was [36], which uses a modified form of evolutionary search. Due to the expense of directly performing discrete search on the search space, this work incurs extremely large computation overhead. Follow-up work by [46] uses the Once-For-All [6] approach to train a supernet for encoder-decoder architectures used in machine translation. Search is performed on subsamples of the supernet that inherit weights to estimate task accuracy. For each target device, the authors train a small neural network regressor on thousands of architectures to estimate latency. As opposed to using a latency estimator, LTS evaluates the latency of each candidate architecture on the target hardware. Notably, by performing the search directly on the target platform, LTS can easily incorporate various hardware performance metrics, e.g., peak memory utilization, for which accurate estimators may not exist. To the best of our knowledge, such holistic integration of multiple hardware metrics in Transformer NAS has not been explored previously.

## 3 Lightweight Transformer Search

We perform an evolutionary search over candidate architectures to extract models that lie on the Pareto-frontier. In contrast to the vast majority of prior methods that deliver a single architecture from the search space, our search is performed over the entire Pareto, generating architectures with a wide range of latency, peak memory utilization, and perplexity with one round of search. This alleviates the need to repeat the NAS algorithm for each hardware performance constraint.

To evaluate candidate models during the search, LTS uses a training-free proxy for the validation perplexity. By incorporating training-free evaluation metrics, LTS, for the first time, performs the entire search directly on the target (constrained) hardware. Therefore, we can use real measurements of hardware performance during the search. Algorithm 1 outlines the iterative process

performed in LTS [3] for finding candidate architectures in the search space ($\mathcal{D}$), that lie on the 3-dimensional Pareto-frontier ($\mathbb{F}$) of perplexity versus latency and memory. At each iteration, a set of points ($\mathbb{F}'$) are subsampled from the current Pareto-frontier. A new batch of architectures ($\mathbb{S}_N$) are then sampled from $\mathbb{F}'$ using evolutionary algorithms ($EA(.)$). The new samples are evaluated in terms of latency ($\mathcal{L}$), peak memory utilization ($\mathcal{M}$), and validation perplexity ($\mathcal{P}$). Latency and memory are measured directly on the target hardware while the perplexity is indirectly estimated using our accurate and training-free proxy methods.

---

**Algorithm 1:** LTS's training-free NAS

**Input:** Search space $\mathcal{D}$, $n_{iter}$
**Output:** Perplexity-latency-memory Pareto-frontier $\mathbb{F}$

1   $\mathcal{L}, \mathcal{M}, \mathcal{P}, \mathbb{F} \leftarrow \emptyset, \emptyset, \emptyset, \emptyset$
2   **while** $N \leq n_{iter}$ **do**
3      $\mathbb{F}' \leftarrow \text{Subsample}(\mathbb{F})$
4      $\mathbb{S}_N \leftarrow EA(\mathbb{F}', \mathcal{D})$
      `// hardware profiling`
5      $\mathcal{L} \leftarrow \mathcal{L} \bigcup \text{Latency}(\mathbb{S}_N)$
6      $\mathcal{M} \leftarrow \mathcal{M} \bigcup \text{Memory}(\mathbb{S}_N)$
      `// estimate perplexity`
7      $\mathcal{P} \leftarrow \mathcal{P} \bigcup \text{Proxy}(\mathbb{S}_N)$
      `// update the Pareto-frontier`
8      $\mathbb{F} \leftarrow \text{LowerConvexHull}(\mathcal{P}, \mathcal{L}, \mathcal{M})$

---

[3]The Pareto-frontier search method in Algorithm 1 is inspired by [13] and [21]. Other possibilities include variations proposed in [20], evaluation of which is orthogonal to our contributions in this work.

Finally, the Pareto-frontier is recalibrated using the lower convex hull of all sampled architectures. In the context of multi-objective NAS, Pareto-frontier points are those where no single metric (e.g., perplexity, latency, and memory) can be improved without degrading at least one other metric [20]. To satisfy application-specific needs, optional upper bounds can be placed on the latency and/or memory of sampled architectures during search.

**Search Space.** Figure 2 shows all elastic parameters in LTS search space, namely, number of layers ($n_{layer}$), number of attention heads ($n_{head}$), decoder output dimension ($d_{model}$), inner dimension of the feed forward network ($d_{inner}$), embedding dimension ($d_{embed}$), and the division factor ($k$) of adaptive embedding [3]. These architectural parameters are compatible with popularly used autoregressive

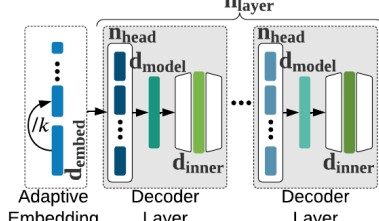

Transformer backbones, e.g., GPT. For preliminaries on autoregressive Transformers, please see Appendix A. We adopt a heterogeneous search space where the backbone parameters are decided on a per-layer basis. This is in contrast to the homogeneous structure commonly used in Transformers [10, 5], which reuses the same configuration for all layers. Compared to homogeneous models, the flexibility associated with heterogeneous architectures enables them to obtain much

Figure 2: Elastic parameters in LTS search space.

better hardware performance under the same perplexity budget (see Section 4.4).

Heterogeneous search space was previously explored in [46]. However, due to the underlying supernet structure, not all design parameters can change freely. As an example, the dimensionality of the *Q, K, V* vectors inside the encoder and decoder layers is fixed to a large value of $512$ to accommodate inheritance from the supernet. Our search space, however, allows exploration of all internal dimensions without constraints. By not relying on the supernet structure, our search space easily encapsulates various Transformer backbones with different configurations of the input/output embedding layers and elastic internal dimensions.

LTS searches over the following values for the architectural parameters in our backbones: $n_{layer} \in \{2, \ldots, 16|1\}$[4], $d_{model} \in \{128, \ldots, 1024|64\}$, $d_{inner} \in \{256, \ldots, 4096|64\}$, and $n_{head} \in \{2, 4, 8\}$. Additionally we explore adaptive input embedding [3] with $d_{embed} \in \{128, 256, 512\}$ and factor $k \in \{1, 2, 4\}$. Once a $d_{model}$ is sampled, we adjust the lower bound of the above range for $d_{inner}$ to $2 \times d_{model}$. Encoding this heuristic inside the search ensures that the acquired models will not suffer from training collapse. Our heterogeneous search space encapsulates more than $10^{54}$ different architectures. Such high dimensionality further validates the critical need for training-free NAS.

### 3.1 Training-free Architecture Ranking

▶ **Low-cost Ranking Proxies.** Recently, Abdelfattah et al. [1] utilize the summation of pruning scores over all model weights as the ranking proxy for Convolutional Neural Networks (CNNs), where a higher score corresponds to higher architecture rank in the search space. White et al. [48] analyze these and more recent proxies and find that no particular proxy performs consistently well over various tasks and baselines, while parameter and floating point operations (FLOPS) count proxies are quite competitive. However, they did not include Transformer-based search spaces in their analysis. To the best of our knowledge, low-cost (pruning-based) proxies have not been evaluated on Transformer search spaces in the language domain. Note that one cannot naively apply these proxies to language models. Specifically, since the embedding layer in Transformers is equivalent to a lookup operation, special care must be taken to omit this layer from the proxy computation. Using this insight, we perform the first systematic study of low-cost proxies for NAS on autoregressive Transformers for text prediction.

We leverage various pruning metrics, namely, `grad_norm`, `snip` [23], `grasp` [45], `fisher` [40], and `synflow` [38]. We also study `jacob_cov` [26] and `relu_log_det` [25] which are low-cost scoring mechanisms proposed for NAS on CNNs in vision tasks. While these low-cost techniques do not perform model training, they require forward and backward passes over the architecture to compute the proxy, which can be time-consuming on low-end hardware. Additionally, the aforesaid pruning techniques, by definition, incorporate the final softmax projection layer in their score assessment. Such an approach seems reasonable for CNNs dealing with a few classification labels,

---

[4] We use the notation $\{v_{min}, \ldots, v_{max}|$step size$\}$ to show the valid range of values.

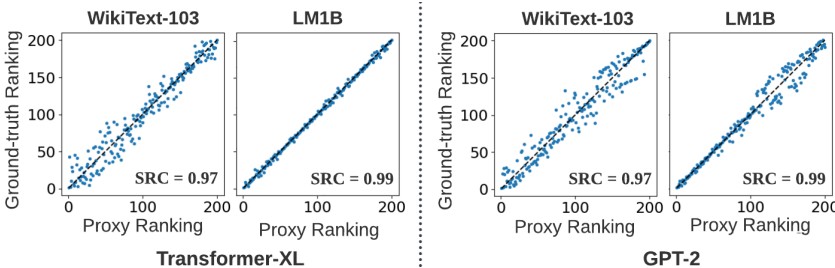

Figure 3: Our training-free zero-cost proxy based on decoder parameter count is highly correlated with the (ground truth) validation perplexity after full training. Each plot contains 200 architectures sampled randomly from the search space of Transformer-XL or GPT-2 backbone.

however, it can skew the evaluation for autoregressive Transformers dealing with a large output vocabulary space. To overcome these shortcomings, we introduce a zero-cost architecture ranking strategy in the next section that outperforms the proposed low-cost proxies in terms of ranking precision, is data free, and does not perform any forward/backward propagation.

▶ **Decoder Parameter Count as a Proxy.** We empirically establish a strong correlation between the parameter count of decoder layers and final model performance in terms of validation perplexity. We evaluate 200 architectures sampled uniformly at random from the search space of two autoregressive Transformer backbones, namely, Transformer-XL and GPT-2. These architectures are trained fully on WikiText-103 and One Billion Word (LM1B) datasets, which consumes over **25000** GPU-hours on NVIDIA A100 and V100 nodes. We compare the ranking obtained using decoder parameter count proxy and the ground truth ranking after full training in Figure 3. On WikiText-103, zero-cost ranking using the decoder parameter count obtains a Spearman's Rank Correlation (SRC) of 0.97 with full training. SRC further increases to 0.99 for the more complex LM1B benchmark on both backbones. This validates that the decoder parameter count is strongly correlated with final model performance, thereby providing a reliable training-free proxy for NAS.

## 4 Experiments

We conduct experiments to seek answers to the following critical questions:

❶ How well can training-free proxies perform compared to training-based methods for estimating the performance of Transformer models?

❷ How does model topology affect the performance of the proposed decoder parameter proxy?

❸ Can our training-free decoder parameter count proxy be integrated inside a search algorithm to estimate the Pareto-frontier? How accurate is such an estimation of the Pareto?

❹ Which models are on the Pareto-frontier of perplexity, latency, and memory for different hardware?

❺ How well do LTS models perform in zero and one-shot settings compared to hand-designed variants when evaluated on downstream tasks?

We empirically answer questions ❶, ❷, ❹, and ❺ in Sections 4.2, 4.3, 4.4, and 4.5 respectively. We further address question ❸ in Appendix C where we show the Pareto-frontier models extracted by the decoder parameter count proxy are very close to the ground truth Pareto-frontier with an average of 0.6% perplexity difference. Additionally, we show the efficacy of the decoder parameter count proxy when performing search on different ranges of model sizes in Appendix C, Figure 12.

### 4.1 Experimental Setup

Please refer to Appendix B for information about the benchmarked datasets, along with details of our training and evaluation setup, hyperparameter optimization, and evolutionary search algorithm.

**Backbones.** We apply our search on two widely used autoregressive Transformer backbones, namely, Transformer-XL [10] and GPT-2 [32] that are trained from scratch with varying architectural hyperparameters. The internal structure of these backbones are quite similar, containing decoder blocks with attention and feed-forward layers. The difference between the backbones lies mainly in their dataflow structure; the Transformer-XL backbone adopts a recurrence methodology over past states coupled with relative positional encoding which enables modeling longer term dependencies.

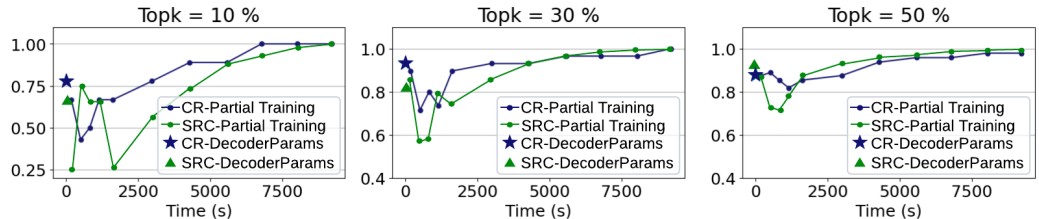

Figure 4: Comparison between partial training and our zero-cost proxy, i.e., decoder parameter count, in terms of ranking performance and timing overhead. Each subplot corresponds to a top$k\%$ of the randomly sampled models, based on their validation perplexity after full training.

**Performance Criteria.** To evaluate the ranking performance of various proxies, we first establish a ground truth ranking of candidate architectures by training them until full convergence. This ground truth ranking is then utilized to compute two performance criteria as follows:

▶ Common Ratio (CR): We define CR as the percentage overlap between the ground truth ranking of architectures versus the ranking obtained from the proxy. CR quantifies the ability of the proxy ranking to identify the top$k\%$ architectures based on their validation perplexity after full training.

▶ Spearman's Rank Correlation (SRC): We use this metric to measure the correlation between the proxy ranking and the ground truth. Ideally, the proxy ranking should have high correlation with the ground truth over the entire search space as well as high-performing candidate models.

### 4.2 How do training-free proxies perform compared to training-based methods?

In this section, we benchmark several proxy methods for estimating the rank of candidate architectures. Specifically, we investigate three different ranking techniques, namely, partial training, low-cost methods, and number of decoder parameters.

▶ **Partial Training.** We first analyze the relationship between validation perplexity after a shortened training period versus that of full training for ranking candidate models. We stop the training after $\tau \in [1.25\%, 87.5\%]$ of the total training iterations needed for model convergence. Figure 4 demonstrates the SRC and CR of partial training with various $\tau$s, evaluated on 100 randomly selected models from the Transformer-XL backbone, trained on WikiText-103. The horizontal axis denotes the average time required for $\tau$ iterations of training across all sampled models. Intuitively, a higher number of training iterations results in a more accurate estimate of the final perplexity. Nevertheless, the increased wall-clock time prohibits training during search and also imposes the need for GPUs. Interestingly, very few training iterations, i.e., $1.25\%$, provide a very good proxy for final performance with an SRC of $> 0.9$ on the entire population. Our training-free proxy, i.e., decoder parameter count, also shows competitive SRC compared to partial training.

▶ **Low-cost Proxies.** We benchmark various low-cost methods introduced in Section 3.1 on 200 randomly sampled architectures from the Transformer-XL backbone, trained on WikiText-103. Figure 5 shows the SRC between low-cost proxies and the ground truth ranking after full training.

We measure the cost of each proxy in terms of FLOPs. As seen, the evaluated low-cost proxies have a strong correlation with the ground truth ranking (even the lowest performing `relu_log_det` has $> 0.8$ SRC), validating the effectiveness of training-free NAS on autoregressive Transformers. The lower performance of `relu_log_det` can be attributed to the much higher frequency of ReLU activations in CNNs, for which the method was originally developed, compared to Transformer-based architectures. Our analysis of randomly selected models with homogeneous structures also shows a strong cor-

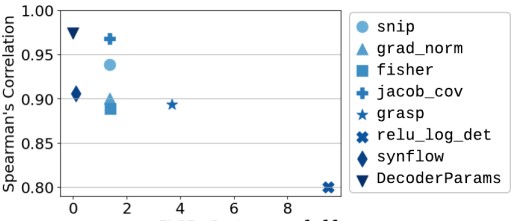

Figure 5: SRC between low-cost proxies and the ground truth ranking after full training of 200 randomly sampled Transformers. The decoder parameter count obtains the best SRC with zero cost.

relation between the low-cost proxies and validation perplexity, with decoder parameter count outperforming other proxies (see Appendix D).

▶ **Parameter Count.** Figure 6a demonstrates the final validation perplexity versus the total number of model parameters for 200 randomly sampled architectures from GPT-2 and Transformer-XL

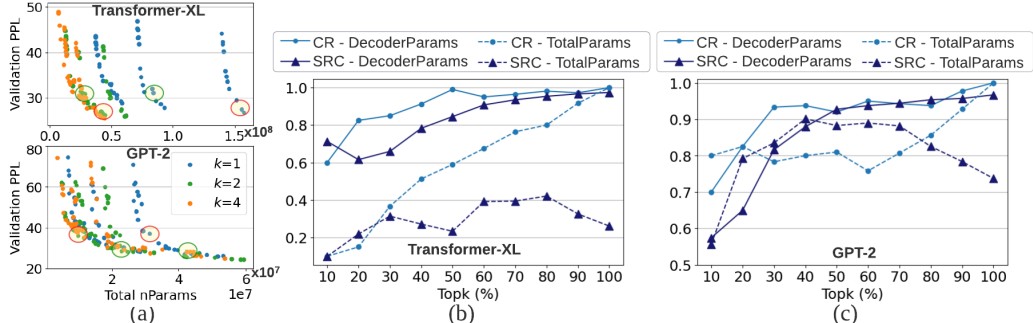

Figure 6: (a) Validation perplexity after full training versus total parameters for 200 randomly sampled architectures trained on WikiText-103. The clear downward trend suggests a strong correlation between parameter count and perplexity. (b), (c) Performance of parameter count proxies for ranking the randomly sampled architectures with Transformer-XL and GPT-2 backbones.

backbones. This figure contains two important observations: (1) the validation perplexity has a downward trend as the number of parameters increases, (2) The discontinuity is caused by the dominance of embedding parameters when moving to the small Transformer regime. We highlight several example points in Figure 6a where the architectures are nearly identical but the adaptive input embedding factor $k$ is changed. Changing $k \in \{1, 2, 4\}$ (shown with different colors in Figure 6a) varies the total parameter count without much influence on the validation perplexity.

The above observations motivate us to evaluate two proxies, i.e., total number of parameters and decoder parameter count. Figures 6b and 6c demonstrate the CR and SRC metrics evaluated on the 200 randomly sampled models divided into top$k$% bins based on their validation perplexity. As shown, the total number of parameters generally has a lower SRC with the validation perplexity, compared to decoder parameter count. This is due to the masking effect of embedding parameters, particularly in the Transformer-XL backbone. The total number of decoder parameters, however, provides a highly accurate, zero-cost proxy with an SRC of $0.97$ with the perplexity over all models, after full training. We further show the high correlation between decoder parameter count and validation perplexity for Transformer architectures with homogeneous decoder blocks in the supplementary material, Appendix D. While our main focus is on autoregressive, decoder-only, Transformers, we provide preliminary results on the ranking performance of parameter count proxies for encoder-only and encoder-decoder Transformers in Appendix J.

### 4.3 How does variation in model topology affect decoder parameter count as a proxy?

The low-cost proxies introduced in Section 3.1, rely on forward and backward passes through the network. As such, they automatically capture the topology of the underlying architecture via the dataflow. The decoder parameter count proxy, however, is topology-agnostic. In this section, we investigate the effect of topology on the performance of decoder parameter count proxy. Specifically, we seek to answer whether for a given decoder parameter count budget, the aspect ratio of the architecture, i.e., trading off the width versus the depth, can affect the final validation perplexity.

We define the aspect ratio of the architecture as $d_{model}$ (=width), divided by $n_{layer}$ (=depth). This metric provides a sense of how skewed the topology is and has been used in prior works which study scaling laws for language models [22]. For a given decoder parameter count budget, we generate several random architectures from the GPT-2 backbone with a wide range of the width-to-depth aspect ratios[5]. The generated models span wide, shallow topologies (e.g., $d_{model}$=1256, $n_{layer}$=2) to narrow, deep topologies (e.g., $d_{model}$=112, $n_{layer}$=100). Figure 7a shows the validation perplexity of said architectures after full training on WikiText-103 versus their aspect ratio. The maximum deviation (from the median) of the validation perplexity is $< 12.8\%$ for a given decoder parameter count, across a wide range of aspect ratios $\in [1, 630]$. Our findings on the heterogeneous search space complement the empirical results by [22] where decoder parameter count largely determines perplexity for homogeneous Transformer architectures, irrespective of shape (see Figure 5 in [22]).

---

[5]We control the aspect ratio by changing the width, i.e., $d_{model}$ while keeping $d_{inner}$=$2 \times d_{model}$ and $n_{head}$=8. The number of layers is then derived such that the total parameter count remains the same.

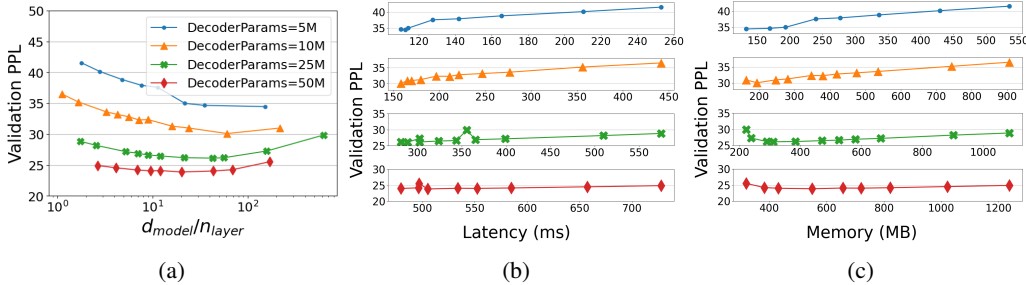

Figure 7: Validation perplexity after full training versus the (a) width-to-depth aspect ratio, (b) latency, and (c) peak memory utilization. Models are randomly generated from the GPT-2 backbone and trained on WikiText-103. For a given decoder parameter count, we observe low variation in perplexity across different models, regardless of their topology. The topology, however, significantly affects the latency (up to $2.8\times$) and peak memory utilization (up to $5.5\times$) for models with the same perplexity.

We observe stable training when scaling models from the GPT-2 backbone up to 100 layers, with the perplexity increasing only when the aspect ratio nears 1. Nevertheless, such deep models are not part of our search space as they have high latency and are unsuitable for lightweight inference. For the purposes of hardware-aware and efficient Transformer NAS, decoder parameter count proxy holds a very high correlation with validation perplexity, regardless of the architecture topology as shown in Figure 7a. We further validate the effect of topology on decoder parameter count proxy for the Transformer-XL backbone in Figure 14 of Appendix E. Our results demonstrate less than 7% deviation (from the median) in validation perplexity for different aspect ratios $\in [8, 323]$.

Note that while models with the same parameter count have very similar validation perplexities, the topology in fact affects their hardware performance, i.e., latency (up to $2.8\times$) and peak memory utilization (up to $5.5\times$), as shown in Figures 7b and 7c. This motivates the need for incorporating hardware metrics in NAS to find the best topology.

## 4.4 Pareto-frontier models for various hardware platforms

We run LTS on different target hardware and obtain a range of Pareto-optimal architectures with various latency/memory/perplexity characteristics. During search, we fix the adaptive input embedding factor to $k = 4$ to search models that are lightweight while ensuring nearly on-par validation perplexity with non-adaptive input embedding. As the baseline Pareto, we benchmark the Transformer-XL (base) and GPT-2 (small) models with homogeneous layers $\in [1, 16]$. This is because the straightforward way to produce architectures of different latency/memory is varying the number of layers (layer-scaling) [42, 10]. We compare our NAS-generated architectures with layer-scaled backbones and achieve better validation perplexity and/or lower latency and peak memory utilization. All baseline[6] and NAS-generated models are trained using the same setup enclosed in Table 2 of Appendix B.

Figure 8 shows the Pareto-frontier architectures found by LTS versus the layer-scaled baseline. Here, all models are trained on the LM1B dataset (See Figure 16 in Appendix G for results on WikiText-103). Note that the Pareto-frontier search is performed in a 3-dimensional space, an example of which is enclosed in Appendix F, Figure 15. For better visualization, in Figure 8 we plot 2-dimensional slices of the Pareto-frontier with validation perplexity on the y-axis and one hardware performance metric (either latency or memory) on the x-axis.

As seen, in the low-latency regime, LTS consistently finds models that have significantly lower perplexity compared to naive scaling of the baseline Transformer-XL or GPT-2. On the Transformer-XL backbone, LTS finds architectures with an average of $19.8\%$ and $28.8\%$ lower latency and memory, while achieving similar perplexity compared to the baseline on ARM CPU. Specifically, the perplexity of the 16-layer Transformer-XL base can be replicated on the ARM device with a lightweight model that is $1.6\times$ faster and utilizes $1.9\times$ less memory during execution. On the Corei7 CPU, the Pareto-frontier models found by LTS are on average $25.8\%$ faster and consume $30.0\%$ less memory under the same validation perplexity constraint. In this setting, LTS finds a model that replicates the perplexity of the 16-layer Transformer-XL base while achieving $1.7\times$ faster runtime and $1.9\times$ less peak memory utilization. The savings are even higher on the GPU device, where

---

[6]The best reported result in the literature for GPT-2 or Transformer-XL might be different based on the specific training hyperparameters, which is orthogonal to our investigation.

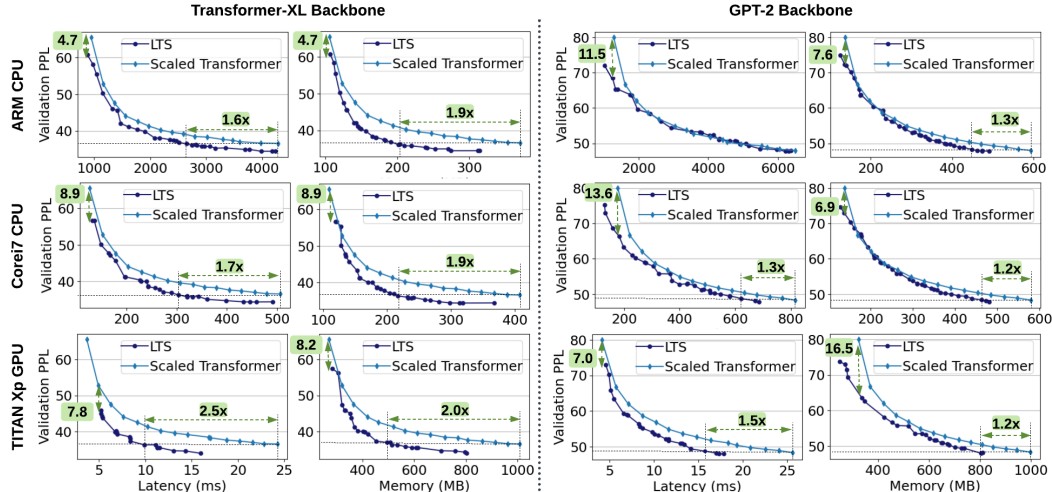

Figure 8: 2D visualization of the perplexity versus latency and memory Pareto-frontier found by LTS, versus the scaled backbone models with varying number of layers, trained on the LM1B dataset. Architectural parameters for models shown here are detailed in Appendix I.

the NAS-generated models achieve the same perplexity as the baseline with average $30.5\%$ lower latency and $27.0\%$ less memory. Specifically, an LTS model with the same perplexity as the 16-layer Transformer-XL base has $2.5\times$ lower latency and consumes $2.0\times$ less peak memory on TITAN Xp.

On the GPT-2 backbone, NAS-generated models consume on average $11.8\%$ less memory while achieving the same validation perplexity and latency on an ARM CPU. The benefits are larger on Corei7 and TitanXP where the latency savings are $13.8\%$ and $11.9\%$, respectively. The peak memory utilization also decreases by $9.7\%$ and $12.9\%$, on average, compared to the baseline GPT-2s on Corei7 and TITAN Xp. Notably, NAS finds new architectures with the same perplexity as the 16-layer GPT-2 with $1.3\times$, $1.5\times$ faster runtime and $1.2\times$ lower memory utilization on Corei7 and TITAN Xp.

Our heterogeneous search space allows us to find a better parameter distribution among decoder layers. Therefore, LTS delivers architectures with better performance in terms of perplexity, while reducing both latency and memory when compared to the homogeneous baselines. We provide the architecture of all baseline and LTS models shown in Figure 8 in Tables 4-7 of Appendix I.

▶ **Search Efficiency.** The main component in LTS search time is the latency/peak memory utilization measurement for candidate architectures since evaluating the model perplexity is instant using the decoder parameter count. Therefore, our search finishes in a few hours on commodity hardware, e.g., taking only 0.9, 2.6, and 17.2 hours on a TITAN Xp GPU, Corei7 CPU, and an ARM core, respectively. To provide more context into the timing analysis, full training of even one 16-layer Transformer-XL base model on LM1B using a machine with $8\times$ NVIDIA V100 GPUs takes 15.8 hours. Once the Pareto-frontier models are identified, the user can pick a model based on their desired hardware constraints and fully train it on the target dataset. LTS is an alternate paradigm to that of training large supernets; our search can run directly on the target device and GPUs are only needed for training the final chosen Pareto-frontier model after search.

In Table 1 we study the ranking performance of partial training (500 steps) versus the decoder parameter count proxy for evaluating 1200 architectures from the Transformer-XL backbone during LTS search. Astonishingly the decoder parameter count proxy gets higher SRC compared to partial training, while effectively removing training from the inner loop of search for NAS.

|  | Train Iter | GPU Hours | $CO_2$e (lbs) | SRC |
|---|---|---|---|---|
| Full Training | 40,000 | 19,024 | 5433 | 1.0 |
| Partial Training | 500 | 231 | 66 | 0.92 |
|  | 5,000 | 2690 | 768 | 0.96 |
| # Decoder Params | **0** | **0** | **∼0** | **0.98** |

Table 1: Ranking abilities of full and partial training versus our proxy for 1200 models sampled during LTS search. Training time is reported for WikiText-103 and NVIDIA V100 GPU. Decoder parameter count proxy obtains an SRC of $0.98$ using zero compute.

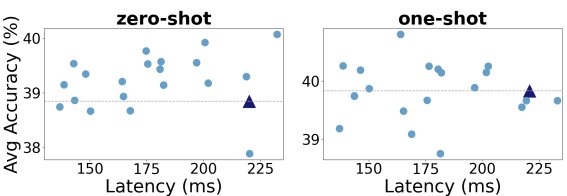

Figure 9: Average zero and one-shot accuracy obtained by LTS models (dots) and the baseline OPT-350M (triangle) across 14 NLP tasks. Latency is measured on an A6000 NVIDIA GPU. Architectural parameters for all models shown here are detailed in Appendix I.

## 4.5 Zero and one-shot performance comparison of LTS models with OPT

Zhang et al. [58] open-source a set of pre-trained decoder-only language models, called OPT, which can be used for zero or few-shot inference on various NLP tasks. Below, we compare the performance of LTS Pareto-frontier models with the hand-designed OPT architecture in zero and one-shot settings.

We use LTS to search for language models with a GPT-2 backbone which have 300M to 500M total parameters to compare with the 350M parameter OPT. Our search space is detailed in Appendix H. The search is conducted with latency as the target hardware metric and decoder parameter count as a proxy for perplexity. Once the search concludes, We train 20 models from the Pareto-frontier along with OPT-350M on 28B tokens from the Pile [17]. The pretrained models are then evaluated on 14 downstream NLP tasks, namely, HellaSwag [57], PIQA [4], ARC (easy and challenge) [9], OpenBookQA [28], WinoGrande [34], and SuperGLUE [44] benchmarks BoolQ, CB, COPA, WIC, WSC, MultiRC, RTE, and ReCoRD. The training hyperparameters and the evaluation setup are outlined in Appendix B. Figure 9 shows the overall average accuracy obtained across all 14 tasks versus the inference latency for LTS models and the baseline OPT. As shown, NAS-generated models achieve a higher average accuracy with lower latency compared to the hand-designed OPT-350M model. We provide a per-task breakdown of zero and one-shot accuracy in Appendix H, Figure 17.

▶ **Zero-shot Performance.** Figure 17a demonstrates the zero-shot accuracy obtained by LTS and OPT-350M on the benchmarked tasks. Compared to the OPT-350M architecture, LTS finds models that achieve higher accuracy and lower latency in the zero-shot setting on all evaluated downstream tasks. Specifically, the maximum achievable accuracy of our NAS-generated models is $0.2 - 8.6\%$ higher than OPT-350M with an average speedup of $1.2\times$. If latency is prioritized, LTS delivers models which are, on average, $1.5\times$ faster and up to $4.6\%$ more accurate than OPT-350M .

▶ **One-shot Performance.** Similar trends can be observed for one-shot evaluation as shown for different tasks in Figure 17b. LTS Pareto-frontier models improve the per-task accuracy of OPT-350M on 12 out of 14 tasks by $0.1 - 8.0\%$, while achieving an average speedup of $1.2\times$. On the same tasks, LTS Pareto-frontier includes models that enjoy up to $1.6\times$ speedup over OPT-350M with an average $1.5\%$ higher accuracy. On the RTE task, the best LTS model has $0.4\%$ lower accuracy but $1.6\times$ faster runtime. On the WSC task, the best performing LTS model obtains a similar one-shot accuracy as OPT-350M, but with $1.5\times$ faster runtime.

## 5   Limitations and Future Work

Decoder parameter count provides a simple yet accurate proxy for ranking autoregressive Transformers. This should serve as a strong baseline for future works on Transformer NAS. Our focus is mainly on autoregressive, decoder-only transformers. We therefore, study perplexity as the commonly used metric for language modeling tasks. Nevertheless, recent literature on scaling laws for Transformers suggest a similar correlation between parameter count and task metrics may exist for encoder only (BERT-style) Transformers or encoder-decoder models used in neural machine translation (NMT) [19]. Additionally, recent findings [39] show specific scaling laws exist between model size and downstream task metrics, e.g., GLUE [43]. Inspired by these observations, we provide preliminary studies that suggest parameter count proxies may be applicable to Transformers in other domains. Detailed investigations of such zero-cost proxies for NAS on heterogeneous BERT-style or NMT models with new performance metrics is an important future avenue of research.

## 6   Acknowledgements

Professor Farinaz Koushanfar's effort has been in parts supported by the NSF TILOS AI institute, award number CCF-2112665.

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
