## A    Preliminaries on Autoregressive Transformers

**Perplexity.** Perplexity is a widely used metric for evaluating the performance of autoregressive language models. This metric encapsulates how well the model can predict a word. Formally, perplexity of a language model $M$ is derived using the entropy formula as:

$$Perplexity(M) = 2^{H(L,M)} = 2^{-\sum_x L(x).log(M(x)))} \tag{1}$$

where $L$ represents the ground truth words. As seen, the perplexity is closely tied with the cross-entropy loss of the model, i.e., $H(L, M)$.

**Parameter count.** Contemporary autoregressive Transformers consist of three main components, namely, the input embedding layer, hidden layers, and the final (softmax) projection layer. The embedding layer often comprises look-up table-based modules that map the input language tokens to vectors. These vectors then enter a stack of multiple hidden layers a.k.a, the decoder blocks. Each decoder block is made up of an attention layer and a feed-forward network. Once the features are extracted by the stack of decoder blocks, the final prediction is generated by passing through the softmax projection layer. When counting the number of parameters in an autoregressive Transformer, the total parameters enclosed in the hidden layers is dubbed the decoder parameter count or equivalently, the non-embedding parameter count. These parameters are architecture-dependent and do not change based on the underlying tokenization or the vocabulary size. The embedding parameter count, however, accounts for the parameters enclosed in the input embedding layer as well as the final softmax projection layer as they are both closely tied to the word embedding and vocabulary size. We visualize an autoregressive Transformer in Figure 10, where the orange blocks contain the decoder parameters and grey blocks hold the embedding parameters.

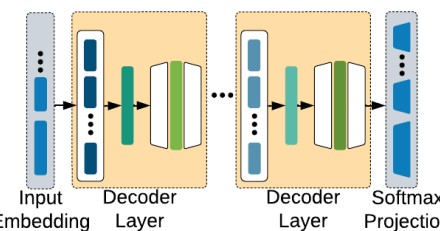

Input Embedding    Decoder Layer    Decoder Layer    Softmax Projection

Figure 10: High-level visualization of different components in autoregressive Transformers. Here, the parameters enclosed in the orange blocks are counted as decoder parameters, while the parameters contained in the gray boxes are included in the embedding parameter count.

## B    Experimental Setup

**Datasets.** We conduct experiments on three datasets, namely, WikiText-103, LM1B, and the Pile. The datasets are tokenized using word-level and byte-pair encoding for models with Transformer-XL and GPT-2 backbones, respectively.

**Training and Evaluation.** We adopt the open-source code by [15] and [30] to implement the GPT-2 and Transformer-XL backbones, respectively. We further use the source code provided in [2] to implement the baseline OPT-350M and LTS models used in zero and one-shot evaluations. Table 2 encloses the hyperparameters used for training. In this paper, each model is trained separately from scratch. In many scenarios, the user only needs to train one model from the Pareto-frontier, which is selected based on their needs for perplexity, latency, and memory. However, if the users are interested in multiple models, they can either train all models separately or fuse them and train them simultaneously using weight sharing as in [46, 55]. As an example, if the user is interested in two particular models from the Pareto-frontier which have 3 and 5 layers, the user can fuse them into a single 5-layer (super)net and train both models at the same time using weight sharing. The cost of training this supernet is roughly the same as training a 5-layer model. Therefore, this simple trick can amortize the training cost for Pareto-frontier models.

Throughout the paper, validation perplexity is measured over a sequence length of 192 and 32 tokens for WikiText-103 and LM1B datasets, respectively. For our zero and one-shot evaluations, we adopt the open-source code by Gao et al. [18]. Inference latency and peak memory utilization are measured on the target hardware for a sequence length of 192, averaged over 10 measurements. The sequence length is increased to 2048 for latency comparison with the OPT baseline. We utilize PyTorch's native benchmarking interface for measuring the latency and memory utilization of candidate architectures.

**Choice of Training Hyperparameters.** For each backbone, dataset, and task, we use the same training setup for all models generated by NAS. This is the common setting used in the vast majority

of NAS papers, including popular benchmarks [12, 35, 56], due to the extremely high cost of NAS combined with hyperparameter optimization (HPO). The setup for our training hyperparameters is based on the evidence provided in prior art in Transformer design [22, 33, 19] and NAS [46, 62, 8]. Specifically, for the range of model sizes studied in this paper, prior work adopts the same batch size (see Table 2.1 in GPT-3 [5]), which suggests there is no significant benefit in optimizing the batch size per architecture. The original GPT-3 paper [5] also adopts the same learning rate scheduler for all models, regardless of their size. Similarly, authors of [22] show that the choice of learning rate scheduler does not have a significant effect on final model performance, which further validates that exploration of the scheduler will not alter the empirical findings in this paper.

Authors of [22] further provide a rule-of-thumb for setting the optimal learning rate (see Equation D.1 of [22]). This rule shows that changes in the optimal learning rate are negligible for the range of model sizes in our search space. We validate this by conducting an experiment that aims to find the optimal learning rate per architecture. We sweep the learning rate $\in [0.0001, 0.001, 0.01, 0.1]$ for 100 randomly sampled models from the GPT-2 backbone and train them on WikiText-103. The studied models span a wide range of configurations with $2-16$ layers and $2-65M$ total parameters. We then pick the optimal learning rate for each architecture, i.e., the one which results in the lowest perplexity. We remeasure the correlation between newly obtained perplexities and the decoder parameter county proxy. Our learning rate optimization experiment results in two important observations: 1) for the vast majority of the architectures ($98\%$), the optimal learning rate is equal to $0.01$, i.e., the value used in all experiments (see Table 2), and 2) the ranking of architectures after convergence remains largely unchanged, leading to a correlation of $0.93$ with decoder parameter count, compared to $0.96$ when using the same learning rate for all models. The above evidence suggests that the same training setup can be used for all architectures in the search space, without affecting the results.

Table 2: LTS training hyperparameters for different backbones. Here, DO represents dropout layers.

| Backbone | Dataset | Tokenizer | # Vocab | Optim. | # Steps | Batch size | LR | Scheduler | Warmup | DO | Attn DO |
|---|---|---|---|---|---|---|---|---|---|---|---|
| Transformer-XL | WT103 | Word | 267735 | LAMB [54] | 4e4 | 256 | 1e-2 | Cosine | 1e3 | 0.1 | 0.0 |
| | LM1B | Word | 267735 | Adam | 1e5 | 224 | 2.5e-4 | Cosine | 2e4 | 0.0 | 0.0 |
| GPT-2 | WT103 | BPE | 50264 | LAMB [54] | 4e4 | 256 | 1e-2 | Cosine | 1e3 | 0.1 | 0.1 |
| | LM1B | BPE | 50264 | LAMB [54] | 1e5 | 224 | 2.5e-4 | Cosine | 2e4 | 0.1 | 0.1 |
| | Pile | BPE | 50272 | Adam | 5.48e4 | 256 | 3e-5 | Linear | 715 | 0.1 | 0.0 |

**Search Setup.** Evolutionary search is performed for 30 iterations with a population size of 100; the parent population accounts for 20 samples out of the total 100; 40 mutated samples are generated per iteration from a mutation probability of 0.3, and 40 samples are created using crossover.

## C   How Good is the Decoder Parameters Proxy for Pareto-frontier Search?

In this Section, we validate whether the decoder parameter count proxy actually helps find Pareto-frontier models which are close to the ground truth Pareto front. We first fully train all 1200 architectures sampled from the Transformer-XL backbone during the evolutionary search (1). Using the validation perplexity obtained after full training, we rank all sampled architectures and extract the ground truth Pareto-frontier of perplexity versus latency. We train the models on the WikiText-103 dataset and benchmark Intel Xeon E5-2690 CPU as our target hardware platform for latency measurement in this experiment.

Figure 11 represents a scatter plot of the validation perplexity (after full training) versus latency for all sampled architectures during the search. The ground truth Pareto-frontier, by definition, is the lower convex hull of the dark navy dots, corresponding to models with the lowest validation perplexity for any given latency constraint. We mark the Pareto-frontier points found by the training-free proxy with orange color. As shown, the architectures that were selected as the Pareto-frontier by the proxy method are either on or very close to the ground truth Pareto-frontier.

We define the mean average perplexity difference as a metric to evaluate the distance ($d_{avg}$) between the proxy and ground truth Pareto-frontier:

$$d_{avg} = \frac{1}{N} \sum_{i=1}^{N} \frac{|p_i - p_{gt,i}|}{p_{gt,i}} \qquad (2)$$

Here, $p_i$ denotes the $i$-th point on the proxy Pareto front and $p_{gt,i}$ is the closest point, in terms of latency, to $p_i$ on the ground truth Pareto front. The mean average perplexity difference for Figure 11

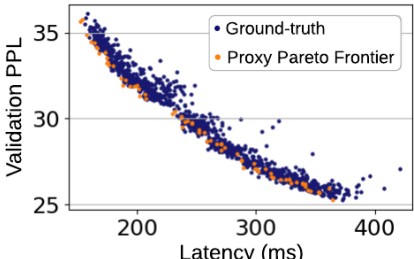

Figure 11: Perplexity versus latency Pareto obtained from full training of $1200$ architectures sampled during NAS on Transformer-XL backbone. Orange points are the Pareto-frontier extracted using decoder parameter count proxy, which lies close to the actual Pareto-frontier. Decoder parameter count holds an SRC of $0.98$ with the ground truth perplexity after full training.

is $d_{avg} = 0.6\%$. This small difference validates the effectiveness of our zero-cost proxy in correctly ranking the sampled architectures and estimating the true Pareto-frontier. In addition to the small distance between the prxoy-estimated Pareto-frontier and the ground truth, our zero-cost proxy holds a high SRC of $0.98$ over the entire Pareto, i.e., all $1200$ sampled architectures.

We further study the decoder parameter proxy in scenarios where the range of model sizes provided for search is limited. We categorize the total $1200$ sampled architectures into different bins based on the decoder parameters. Figure 12 demonstrates the SRC between the decoder parameter count proxy and the validation perplexity after full training for different model sizes. The proposed proxy provides a highly accurate ranking of candidate architectures even when exploring a small range of model sizes.

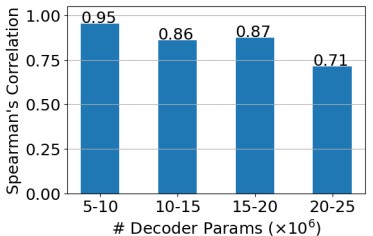

Figure 12: SRC between the decoder parameter count proxy and validation perplexity. Results are gathered on $1200$ models grouped into four bins based on their decoder parameter count. Our proxy performs well even when exploring within a small range of model sizes.

## D  Analysis on Homogeneous Models

In this section, we evaluate the efficacy of the proposed proxies on the homogeneous search space, i.e., when all decoder layers have the same parameter configuration. In this scenario, the parameters are sampled from the valid ranges in Section 3 to construct one decoder block. This block is then replicated based on the selected $n_{layer}$ to create the Transformer architecture. In what follows, we provide experimental results gathered on $100$ randomly sampled Transformer models from the Transformer-XL backbone with homogeneous decoder blocks, trained on WikiText-103.

▶ **Low-cost Proxies.** Figure 13a demonstrates the SRC between various low-cost methods and the validation perplexity after full training. On the horizontal axis, we report the total computation required for each proxy in terms of FLOPs. Commensurate with the findings on the heterogeneous models, we observe a strong correlation between the low-cost proxies and validation perplexity, with the decoder parameter count outperforming other proxies. Note that we omit the `relu_log_det` method from Figure 13a as it provides a low SRC of $0.42$ due to heavy reliance on ReLU activations.

▶ **Parameter Count.** As seen in Figure 13b, the total parameter count has a low SRC with the validation perplexity while the decoder parameter count provides an accurate proxy with an SRC of $0.95$ over all architectures. These findings on the homogeneous search space are well-aligned with the observations in the heterogeneous space.

## E  How Does Model Topology Affect the Training-free Proxies?

Figure 14a shows the validation perplexity versus the aspect ratio of random architectures sampled from the Transformer-XL backbone and trained on WikiText-103. Here, the models span wide, shallow topologies (e.g., $d_{model}$=1024, $n_{layer}$=3) to narrow, deep topologies (e.g., $d_{model}$=128, $n_{layer}$=35). The maximum change in the validation perplexity for a given decoder parameter count is $< 7\%$ for a wide range of aspect ratios $\in [8, 323]$. Nevertheless, for the same decoder parameter count budget, the latency and peak memory utilization vary by $1.3\times$ and $2.0\times$ as shown in Figures 14b and 14c.

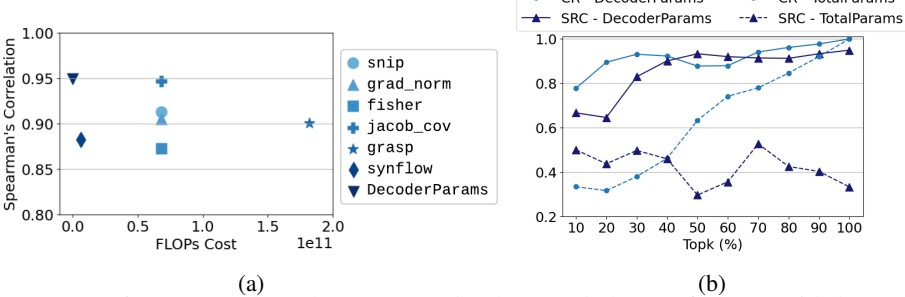

(a)                                                    (b)

Figure 13: Experiments conducted on 100 randomly sampled Transformers with homogeneous decoder blocks, trained on WikiText-103. (a) SRC between the ranking obtained from low-cost proxies and the ground truth ranking after full training. The decoder parameter count obtains the best SRC with zero cost. (b) Performance of parameter count proxies. The decoder parameter count provides a very accurate ranking proxy with an SRC of $0.95$ over all models.

For deeper architectures (more than $40$ layers) with the Transformer-XL backbone, we observe an increase in the validation perplexity, which results in a deviation from the pattern in Figure 14a. This observation is associated with the inherent difficulty in training deeper architectures, which can be mitigated with the proposed techniques in the literature [47]. Nevertheless, such deep models have a high latency, which makes them unsuitable for lightweight inference. For hardware-aware and efficient Transformer NAS, our search space contains architectures with less than $16$ layers. In this scenario, the decoder parameter count proxy holds a very high correlation with validation perplexity, regardless of the architecture topology as shown in Figure 14a.

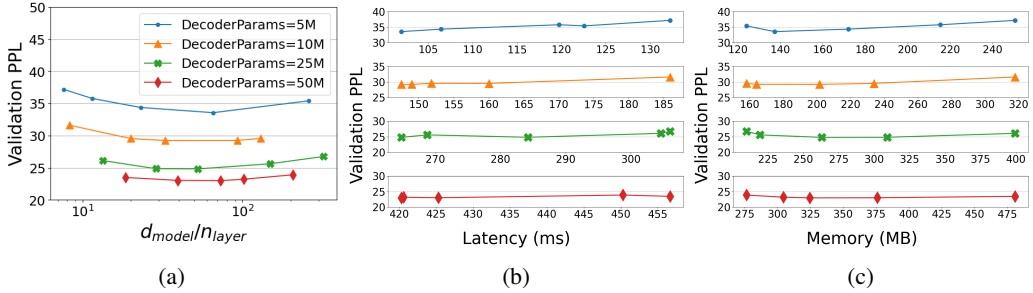

(a)                                    (b)                                    (c)

Figure 14: Validation perplexity after full training versus (a) the width-to-depth aspect ratio, (b) latency, and (c) peak memory utilization. Models are randomly generated from the Transformer-XL backbone and trained on WikiText-103. For a given decoder parameter count, we observe low variation in perplexity across different models, regardless of their topology. The topology, however, significantly affects the latency and peak memory utilization for models with the same perplexity.

## F    3D Pareto Visualization

Figure 15 visualizes the 3-dimensional Pareto obtained during search on the GPT-2 backbone. Here, the black and blue points denote regular and Pareto-frontier architectures, respectively. The pair of red dots are architectures which match in both memory and decoder parameter count ($\sim$ perplexity). However, as shown, their latency differs by $2\times$. The pair of green points correspond to models with the same decoder parameter count ($\sim$ perplexity) and latency, while the memory still differs by 30MB, which is non-negligible for memory-constrained application. In a 2-objective Pareto-frontier search of perplexity versus memory (or latency), each pair of red (or green) dots will result in similar evaluations. While in reality, they have very different characteristics in terms of the overlooked metric. This experiment validates the need for multi-objective Pareto-frontier search, which simultaneously takes into account multiple hardware performance metrics.

## G    LTS Pareto-frontier on WikiText-103

We compare the Pareto-frontier architectures found by LTS with the baseline after full training on the WikiText-103 dataset in Figure 16. Commensurate with the findings on the LM1B dataset, the

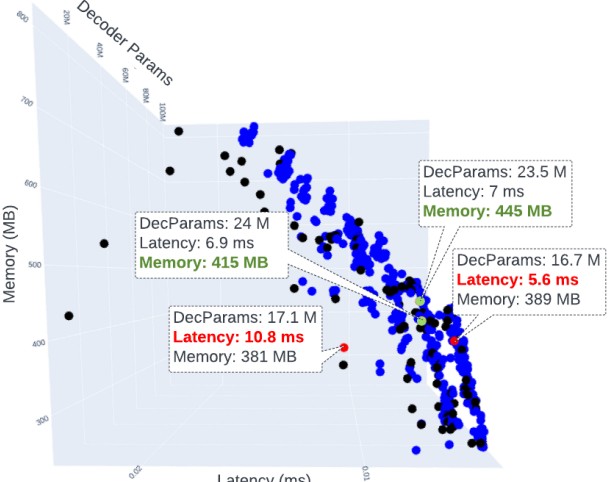

Figure 15: 3D visualization of our multi-objective NAS for the GPT-2 backbone on TITAN Xp GPU. Architectures with similar memory and decoder parameter count can result in drastically different runtimes (up to $2\times$ difference). Similarly, architectures with similar decoder parameter count and latency may have different peak memory utilization. Therefore, it is important to perform multi-objective NAS where several hardware characteristics are simultaneously taken into account when extracting the Pareto-frontier.

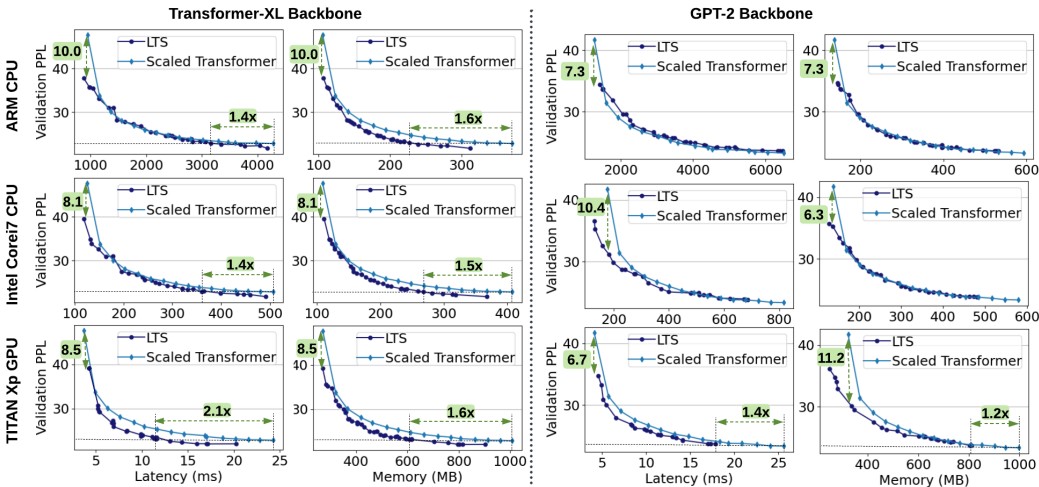

Figure 16: 2D visualization of the perplexity versus latency and memory Pareto-frontier found by LTS and scaled backbone models with varying number of layers. All models are trained on the WikiText-103 dataset. The architectural parameters for all models are enclosed in Appendix I.

NAS-generated models outperform the baselines in at least one of the three metrics, i.e., perplexity, latency, and peak memory utilization. We note that the gap between the baseline models and those obtained from NAS is larger when training on the LM1B dataset. This is due to the challenging nature of LM1B, which exceeds the WikiText-103 dataset size by $\sim 10\times$. Thus, it is harder for hand-crafted baseline models to compete with the optimized LTS architectures on LM1B.

On the Transformer-XL backbone, the models on LTS Pareto-frontier for the ARM CPU have, on average, $3.8\%$ faster runtime and $20.7\%$ less memory under the same validation perplexity budget. On the Corei7, the runtime and memory savings increase to $13.2\%$ and $19.6\%$, respectively, while matching the baseline perplexity. We achieve our highest benefits on TITAN Xp GPU where LTS Pareto-frontier models have, on average, $31.8\%$ lower latency and $21.5\%$ lower memory utilization. Notably, the validation perplexity of the baseline 16-layer Transformer-XL base can be achieved with a lightweight model with $2.1\times$ lower latency while consuming $1.6\times$ less memory at runtime.

On the GPT-2 backbone, LTS achieves $6.3 - 11.2$ lower perplexity in the low-latency-and-memory regime. As we transition to larger models and higher latency, our results show that the GPT-2 architecture is nearly optimal on WikiText-103 when performing inference on a CPU. The benefits are more significant when targeting a GPU; For any given perplexity achieved by the baseline, LTS Pareto-frontier on TITAN Xp delivers, on average, $9.0\%$ lower latency and $4.5\%$ lower memory. Therefore, the perplexity and memory of the baseline 16-layer GPT-2 can be achieved by a new model that runs $1.4\times$ faster and consumes $1.2\times$ less memory on TITAN Xp.

# H   Zero and One-Shot Evaluation of LTS Models

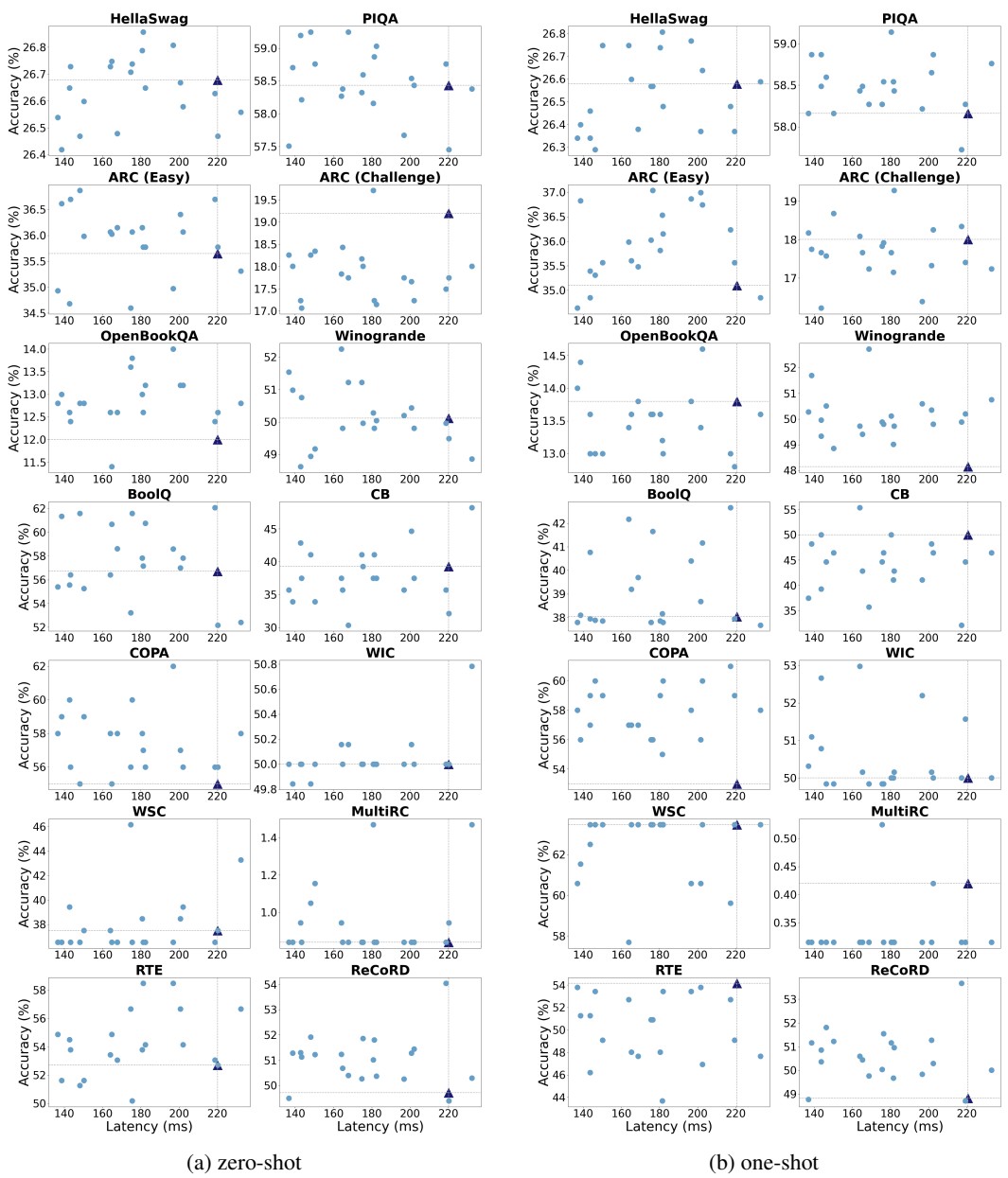

(a) zero-shot                                    (b) one-shot

Figure 17: LTS Pareto-frontier models (dots) achieve a higher zero and one-shot accuracy with lower latency compared to the hand-designed OPT-350M model (triangle). Latency is measured on an A6000 NVIDIA GPU. Architectural parameters for all models shown here are detailed in Appendix I.

For this experiment, we design our search space to cover models with a similar parameter count budget as the OPT-350M model. To this end, we search over the following values for the architectural parameters: $n_{layer} \in \{3, \dots, 29|1\}$, $d_{model} \in \{512, \dots, 1472|64\}$, $d_{inner} \in \{512, \dots, 6080|64\}$, and $n_{head} \in \{2, 4, 8, 16\}$. To directly compare with OPT, we use a generic, non-adaptive embedding layer for our models. Therefore, the search space does not include the $k$ factor and $d_{embed}$=$d_{model}$.

Figures 17a and 17b show the per-task zero and one-shot performance of LTS models and OPT-350M. Please refer to Section 4.5 of the main paper for a summarization of the results in these figures.

# I Architecture Details

Tables 4, 5, 6, 7 enclose the architecture parameters for the baseline and NAS-generated models in Figures 8 and 16 for Transformer-XL and GPT-2 backbones. Table 3 further holds the architecture details of models used in our zero and one-shot evaluations of Figures 9 and 17. For each target hardware, the rows of the table are ordered based on increasing decoder parameter count (decreasing validation perplexity). For all models, $d_{head}=d_{model}/n_{head}$ and $d_{embed}=d_{model}$. For models in Tables 4, 5, 6, 7, the adaptive input embedding factor is set to $k = 4$. The models in Table 3, however, use the generic, non-adaptive, input embedding ($k = 1$) following the original OPT architecture [58].

# J Transformers in other Domains

In what follows, we perform preliminary experiments on Transformers used on other domains to investigate the applicability of parameter-based proxies for ranking.

**Encoder-only Transformers.** BERT [11] is a widely popular Transformer composed of encoder blocks, which is used in a variety of tasks, e.g., question answering and language inference. The main difference between BERT and the Transformers studied in this paper is the usage of bidirectional versus causal attention. Specifically, the encoder blocks in BERT are trained to compute attention between each input token and all surrounding tokens. In autoregressive models, however, attention is only computed for tokens appearing prior to the current token. BERT is trained with a mixture of masked language modeling and next sentence prediction objectives to ensure applicability to language modeling as well as downstream language understanding tasks. We use the architectural parameters described in Section 3 to construct the search space and randomly sample 300 models from the BERT backbone. We then train all models on WikiText-103 for 40K steps following the training setup provided in the original BERT paper [11] for the batch size, sequence length, optimizer, learning rate, vocabulary size, and tokenizer. Figure 18 demonstrates the CR and SRC of encoder parameter count and test perplexity measured on various top$k\%$ performing BERT models. As seen, both the encoder and total parameter count provide a highly accurate proxy for test perplexity of BERT, achieving an SRC of $0.96$ and $0.98$, respectively. This trend suggests that parameter-based proxies for NAS can be applicable to encoder-only search spaces as well.

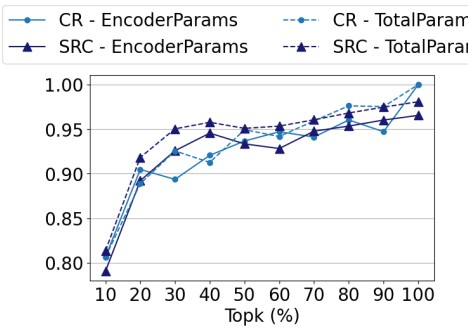

Figure 18: Performance of parameter count proxies on 300 randomly sampled models from the BERT backbone, trained on WikiText-103. Both encoder and total parameter counts provide a very accurate ranking proxy with an SRC of 0.96 and 0.98 over all models, respectively.

**Encoder-Decoder Transformers.** Transformers in this domain comprise both encoder and decoder layers with bidirectional and causal attention computation. This unique structure makes these models suitable for sequence-to-sequence tasks such as Neural Machine Translation (NMT). Recent work [19] shows that the performance of encoder-decoder Transformers also follows a scaling law with model size. This power-law behavior between model size and performance can be leveraged to develop training-free proxies for ranking these architectures during search. We test our hypothesis by performing experiments on the open-source NMT benchmark by [60, 59] which consists of 2000 Transformers trained on various language pairs. The pre-trained Transformers in this benchmark have homogeneous layers, i.e., the architectural parameters are the same for all layers and identical for the encoder and the decoder. In addition to architectural parameters, the search space for this benchmark also includes various BPE tokenization and learning rates. We, therefore, pre-process the benchmark by gathering all instances of Transformers for a fixed BPE. Then for each given architecture, we keep the results corresponding to the best-performing learning rate.

Figure 19 shows a heatmap of the SRC between parameter count proxies and perplexity as well as the BLEU score. As seen, the ranking performance of total parameter count versus non-embedding

parameter count, i.e., parameters enclosed in the encoder and decoder blocks, is largely similar. On certain tasks, e.g., 'ja-en', 'so-en', and 'sw-en' the parameter count proxies perform quite well, achieving a high SRC with both the BLEU score and perplexity. Interestingly, on 'so-en' and 'sw-en', the parameter count and performance are inversely correlated, which may be due to the limited training data for these language pairs which gives smaller models a leading advantage over larger architectures. While these preliminary results show promise for parameter-based proxies in NAS for NMT, several aspects require further investigation, e.g., the effect of architectural heterogeneity and dataset size on the performance of these proxies. Studying these aspects may perhaps lead to a new formulation of training-free proxies for NMT and are out of scope for this paper.

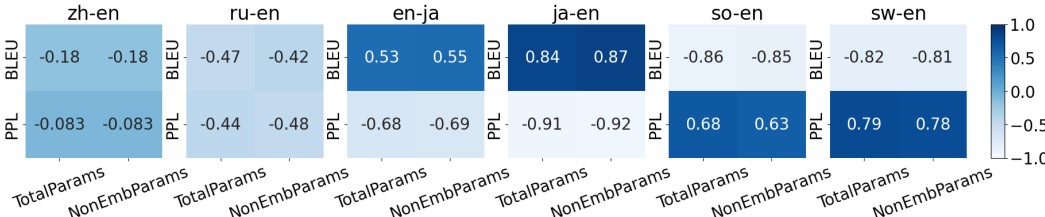

Figure 19: SRC between parameter count proxies and performance metrics, i.e., BLEU score and perplexity (PPL) for translation between various language pairs. The "NonEmbParams" label denotes the parameters enclosed in the encoder and decoder blocks while the "TotalParams" label corresponds to the total parameter count including those in the embedding layers. Here, darker versus lighter colors show a high positive and negative correlation, respectively.

## K    Ethics Statement and Broader Impact

We provide an extremely lightweight method for NAS on autoregressive Transformers. Our work is likely to increase the adoption of NAS in the NLP domain, providing several prevalent benefits:

Firstly, more widespread adoption of automated techniques, e.g., NAS eliminates the need for laborious trials and error for manual design of Transformer architectures, freeing up hundreds of hours of man-power as well as computational resources. Secondly, automating architecture design can trigger the generation of new models with superior performance, which benefits the ever-growing applications of NLP in the everyday life. Finally, by making the search algorithm efficient, we ensure it can be accessible to the general scientific public without need for any expensive mode training, thereby minimizing the unwanted byproducts of the Deep Learning era such as the carbon footprint, and power consumption. While the benefits of automation in NLP are plenty, it can lead to potential side effects that have not been yet fully unveiled. Since our work advances the use of NAS in the NLP design pipeline, there is need for scrutiny of the models which have been automatically designed with respect to aspects such as bias, misinformation, and nefarious activity, to name a few.

Table 3: Detailed architectural parameters for all models in Figure 9 with GPT-2 backbone.

|  | $n_{layer}$ | $d_{model}$ | $n_{head}$ | $d_{inner}$ | DecoderParams (M) |
|---|---|---|---|---|---|
| baseline (OPT-350M) | 24 | 1024 | 16 | 4096 | 304.4 |
| M1 | 26 | 1024 | 16 | 2816 | 261.4 |
| M2 | 15 | 1280 | 16 | 4480 | 273.2 |
| M3 | 24 | 1280 | 8 | 1856 | 274.3 |
| M4 | 16 | 1344 | 8 | 3840 | 283.8 |
| M5 | 14 | 1344 | 8 | 4800 | 284.8 |
| M6 | 20 | 1216 | 4 | 3456 | 289.2 |
| M7 | 16 | 1344 | 16 | 4096 | 294.8 |
| M8 | 28 | 1344 | 8 | 1344 | 306.6 |
| M9 | 28 | 1088 | 8 | 2816 | 306.7 |
| M10 | 26 | 1152 | 16 | 2816 | 309.4 |
| M11 | 25 | 832 | 2 | 5760 | 310.9 |
| M12 | 20 | 1280 | 16 | 3456 | 310.9 |
| M13 | 19 | 1280 | 8 | 3840 | 314.2 |
| M14 | 26 | 1152 | 4 | 3008 | 320.9 |
| M15 | 19 | 1472 | 8 | 2816 | 325.5 |
| M16 | 13 | 1472 | 4 | 5568 | 329.0 |
| M17 | 14 | 1480 | 2 | 5824 | 367.3 |
| M18 | 20 | 1152 | 8 | 5760 | 374.3 |
| M19 | 26 | 1024 | 4 | 5696 | 414.8 |
| M20 | 25 | 1408 | 8 | 3136 | 422.3 |

Table 4: Detailed architectural parameters for all models in Figure 8 with Transformer-XL backbone.

| | | $n_{layer}$ | $d_{model}$ | $n_{head}$ | $d_{inner}$ | DecoderParams (M) |
|---|---|---|---|---|---|---|
| | baseline | $\in[1,16]$ | 512 | 8 | 2048 | - |
| **ARM** | M1 | 2 | 512 | [2, 2] | [1216, 1280] | 5.2 |
| | M2 | 3 | 320 | [2, 4, 2] | [1472, 2368, 3392] | 6.2 |
| | M3 | 2 | 512 | [2, 2] | [2560, 2176] | 7.5 |
| | M4 | 2 | 512 | [2, 2] | [3904, 1792] | 8.5 |
| | M5 | 2 | 640 | [2, 2] | [3520, 3456] | 13.0 |
| | M6 | 2 | 704 | [8, 2] | [3904, 3968] | 16.1 |
| | M7 | 2 | 832 | [2, 2] | [3264, 3968] | 19.0 |
| | M8 | 2 | 960 | [2, 2] | [3648, 3968] | 23.9 |
| | M9 | 2 | 960 | [2, 2] | [3904, 3968] | 24.4 |
| | M10 | 3 | 960 | [2, 2, 2] | [1856, 2368, 3392] | 28.5 |
| | M11 | 3 | 832 | [2, 2, 2] | [3904, 3968, 3008] | 28.5 |
| | M12 | 3 | 960 | [2, 4, 2] | [3328, 2368, 3200] | 30.9 |
| | M13 | 3 | 960 | [4, 2, 2] | [3648, 3584, 3584] | 34.6 |
| | M14 | 3 | 960 | [2, 2, 2] | [3904, 3584, 3456] | 34.9 |
| | M15 | 3 | 960 | [2, 2, 8] | [4032, 3968, 3904] | 36.7 |
| | M16 | 4 | 896 | [4, 2, 8, 2] | [3904, 3008, 3520, 3584] | 41.2 |
| | M17 | 4 | 960 | [8, 8, 8, 4] | [4032, 3968, 2880, 3200] | 45.5 |
| | M18 | 4 | 960 | [2, 2, 2, 2] | [3840, 3904, 3520, 3072] | 46.0 |
| | M19 | 4 | 960 | [2, 2, 2, 2] | [4032, 3648, 3136, 4032] | 47.0 |
| | M20 | 4 | 960 | [8, 2, 4, 8] | [4032, 3584, 3840, 3584] | 47.4 |
| | M21 | 4 | 960 | [2, 2, 4, 2] | [3904, 3968, 3840, 3584] | 47.8 |
| | M22 | 5 | 960 | [2, 2, 2, 2, 2] | [3904, 3968, 3264, 3456, 3200] | 57.3 |
| | M23 | 5 | 960 | [2, 2, 2, 8, 2] | [3904, 3648, 3136, 3648, 3840] | 58.0 |
| | M24 | 6 | 960 | [2, 2, 2, 2, 8] | [3328, 2624, 3392, 2944, 3008, 3904] | 64.6 |
| | M25 | 6 | 960 | [2, 2, 4, 2, 8, 8] | [3584, 2624, 3392, 3968, 3008, 3328] | 65.9 |
| | M26 | 6 | 960 | [2, 4, 2, 2, 2, 2] | [2112, 3840, 3328, 3264, 3968, 3648] | 66.4 |
| | M27 | 6 | 960 | [2, 4, 2, 2, 8, 2] | [3904, 3008, 3392, 3648, 3392, 3584] | 67.9 |
| | M28 | 6 | 960 | [2, 2, 2, 2, 2, 4] | [3968, 3968, 3456, 3456, 3776, 2432] | 68.1 |
| | M29 | 6 | 960 | [2, 4, 8, 4, 2, 8] | [3904, 3008, 3392, 3200, 3968, 3904] | 68.8 |
| | M30 | 6 | 960 | [8, 8, 2, 4, 2, 4] | [3904, 3648, 3136, 3648, 3200, 3840] | 68.8 |
| | M31 | 6 | 960 | [8, 4, 8, 4, 2, 8] | [3904, 3648, 3392, 3200, 3968, 3840] | 69.9 |
| | M32 | 8 | 896 | [4, 2, 2, 4, 4, 2, 4, 8] | [3584, 3968, 3392, 3904, 2240, 1856, 2560, 3264] | 76.6 |
| | M33 | 8 | 896 | [4, 2, 2, 2, 4, 4, 4, 2] | [3584, 3584, 3520, 2368, 2752, 4032, 3520, 3264] | 79.9 |
| | M34 | 9 | 896 | [4, 2, 4, 4, 8, 2, 8, 8, 2] | [3840, 3136, 3520, 2880, 3200, 3008, 3328, 2560, 3136] | 87.5 |
| | M35 | 8 | 960 | [2, 4, 4, 4, 4, 8, 2] | [3968, 3584, 3520, 3072, 3968, 4032, 1856, 3712] | 90.2 |
| | M36 | 12 | 832 | [2, 4, 4, 2, 2, 8, 8, 8, 4, 4, 2, 8] | [3136, 2112, 2112, 2368, 2752, 2432, 2432, 2176, 3456, 3712, 2880, 3712] | 97.0 |
| | M37 | 9 | 960 | [4, 4, 8, 2, 2, 2, 8, 8, 2] | [2112, 3008, 3520, 3648, 3968, 4032, 1984, 3200, 3520] | 97.2 |
| | M38 | 9 | 960 | [8, 2, 4, 2, 8, 8, 8, 2, 2] | [3968, 3008, 3520, 3200, 3200, 4032, 1984, 2816, 3520] | 97.7 |
| | M39 | 12 | 832 | [4, 4, 4, 2, 2, 8, 4, 8, 2, 8, 2, 8] | [3136, 3968, 2112, 2368, 3072, 2240, 2624, 2112, 3456, 3072, 2880, 3264] | 98.7 |
| **Corei7** | M1 | 2 | 384 | [2, 2] | [896, 2816] | 4.3 |
| | M2 | 2 | 576 | [2, 2] | [1792, 2816] | 8.6 |
| | M3 | 2 | 576 | [2, 2] | [1408, 3776] | 9.3 |
| | M4 | 2 | 832 | [2, 2] | [1728, 1536] | 12.4 |
| | M5 | 2 | 768 | [2, 2] | [3776, 1920] | 14.7 |
| | M6 | 2 | 768 | [2, 2] | [2112, 3584] | 14.7 |
| | M7 | 2 | 832 | [2, 2] | [3776, 3392] | 18.9 |
| | M8 | 2 | 832 | [2, 2] | [3968, 3584] | 19.5 |
| | M9 | 2 | 960 | [2, 4] | [1984, 3840] | 20.4 |
| | M10 | 2 | 960 | [8, 8] | [3968, 3584] | 23.7 |
| | M11 | 2 | 960 | [2, 2] | [3904, 3904] | 24.2 |
| | M12 | 3 | 896 | [2, 2, 2] | [2304, 3904, 3904] | 30.2 |
| | M13 | 3 | 960 | [2, 2, 4] | [2176, 3840, 2880] | 30.9 |
| | M14 | 3 | 960 | [2, 2, 4] | [3776, 2880, 3904] | 34.1 |
| | M15 | 3 | 960 | [2, 8, 2] | [3840, 3840, 3904] | 36.1 |
| | M16 | 3 | 960 | [2, 8, 8] | [3904, 3840, 3904] | 36.2 |
| | M17 | 3 | 960 | [2, 2, 8] | [3968, 3904, 3904] | 36.5 |
| | M18 | 4 | 960 | [2, 4, 2, 2] | [3904, 2112, 4032, 3584] | 44.6 |
| | M19 | 4 | 960 | [2, 2, 2, 4] | [2112, 3840, 3904, 3904] | 44.9 |
| | M20 | 4 | 960 | [2, 4, 8, 4] | [3776, 3392, 3520, 3904] | 46.5 |
| | M21 | 4 | 960 | [2, 2, 2, 4] | [3904, 3776, 3904, 3904] | 48.2 |
| | M22 | 5 | 960 | [2, 2, 2, 2, 2] | [3776, 1984, 3904, 3904, 3456] | 55.8 |
| | M23 | 5 | 960 | [2, 4, 2, 4, 2] | [3968, 3584, 3520, 3904, 3200] | 58.0 |
| | M24 | 5 | 960 | [2, 4, 4, 4, 2] | [3776, 3840, 3904, 3904, 3968] | 60.3 |
| | M25 | 6 | 960 | [2, 4, 4, 2, 2, 4] | [3776, 3840, 3904, 3904, 3008, 2304] | 67.5 |
| | M26 | 6 | 960 | [2, 4, 2, 4, 2, 4] | [3776, 2112, 4032, 3584, 3200, 4032] | 67.5 |
| | M27 | 6 | 960 | [2, 4, 2, 4, 4, 4] | [3776, 3840, 3904, 4032, 3648, 2432] | 69.2 |
| | M28 | 6 | 960 | [4, 2, 8, 4, 2, 2] | [3840, 3712, 3520, 4032, 3200, 4032] | 70.6 |
| | M29 | 7 | 960 | [2, 2, 8, 4, 2, 2, 4] | [3776, 3840, 3904, 1856, 3072, 3648, 4032] | 78.7 |
| | M30 | 8 | 960 | [2, 2, 2, 4, 2, 4, 8, 2] | [3392, 1792, 3904, 3904, 3200, 2432, 1792, 2496] | 80.9 |
| | M31 | 8 | 960 | [2, 4, 4, 2, 4, 8, 4, 4] | [3776, 3008, 4032, 3904, 3520, 3136, 1984, 3648] | 88.8 |
| | M32 | 8 | 960 | [8, 2, 4, 8, 4, 4, 4, 8] | [3776, 3008, 3904, 3904, 2176, 4032, 4032, 3648] | 91.6 |
| | M33 | 13 | 768 | [2, 8, 2, 4, 2, 2, 4, 2, 2, 8, 8, 8, 4] | [3776, 2112, 1600, 3904, 3840, 2880, 2304, 3200, 2048, 2944, 2816, 3328, 3968] | 97.9 |
| | M34 | 9 | 960 | [4, 2, 4, 4, 4, 4, 8, 8, 2] | [3840, 3136, 3520, 4032, 3200, 4032, 3648, 2112, 2368] | 98.9 |
| | M35 | 9 | 960 | [8, 2, 8, 8, 2, 4, 8, 2, 2] | [3520, 3008, 2880, 4032, 3200, 2432, 4032, 3904, 3136] | 99.4 |

Table 5: Detailed architectural parameters for all models in Figure 8 with Transformer-XL backbone.

| | $n_{layer}$ | $d_{model}$ | $n_{head}$ | $d_{inner}$ | DecoderParams (M) |
|---|---|---|---|---|---|
| baseline | ∈[1,16] | 512 | 8 | 2048 | - |
| M1 | 2 | 384 | [2, 2] | [1152, 2432] | 4.2 |
| M2 | 2 | 448 | [8, 2] | [2944, 3008] | 7.4 |
| M3 | 2 | 576 | [2, 2] | [2048, 1728] | 7.7 |
| M4 | 2 | 512 | [2, 2] | [2368, 3072] | 8.2 |
| M5 | 2 | 832 | [8, 2] | [3264, 3072] | 17.5 |
| M6 | 2 | 768 | [2, 2] | [3968, 4032] | 18.2 |
| M7 | 2 | 896 | [8, 4] | [4032, 2880] | 20.4 |
| M8 | 2 | 960 | [4, 8] | [3968, 3008] | 22.6 |
| M9 | 2 | 960 | [4, 8] | [3968, 3648] | 23.9 |
| M10 | 2 | 960 | [2, 2] | [3840, 3968] | 24.2 |
| M11 | 3 | 896 | [8, 4, 8] | [4032, 2112, 3392] | 29.2 |
| M12 | 3 | 896 | [2, 2, 2] | [3840, 2880, 3840] | 31.0 |
| M13 | 3 | 960 | [2, 2, 2] | [3584, 3072, 2624] | 31.7 |
| M14 | 3 | 960 | [4, 2, 2] | [3840, 3008, 3840] | 34.4 |
| M15 | 3 | 960 | [8, 2, 8] | [4032, 4032, 3520] | 36.1 |
| M16 | 3 | 960 | [2, 2, 8] | [3584, 4032, 4032] | 36.2 |
| M17 | 3 | 960 | [2, 2, 8] | [4032, 4032, 3840] | 36.7 |
| M18 | 3 | 960 | [8, 4, 8] | [4032, 4032, 4032] | 37.1 |
| M19 | 4 | 896 | [4, 4, 8, 8] | [4032, 3456, 3328, 3392] | 41.6 |
| M20 | 4 | 960 | [4, 2, 8, 8] | [3840, 3008, 3328, 3584] | 44.9 |
| M21 | 4 | 960 | [2, 2, 8, 8] | [4032, 3968, 3904, 3840] | 48.7 |
| M22 | 4 | 960 | [4, 2, 4, 4] | [3840, 4032, 3904, 4032] | 48.8 |
| M23 | 5 | 960 | [4, 2, 4, 4, 8] | [3840, 3008, 3392, 2496, 4032] | 55.3 |
| M24 | 5 | 960 | [4, 2, 8, 8, 8] | [3840, 3008, 3840, 3328, 3968] | 57.6 |
| M25 | 5 | 960 | [2, 2, 4, 4, 4] | [3968, 4032, 3328, 4032, 2752] | 57.9 |
| M26 | 6 | 896 | [8, 4, 8, 4, 8, 8] | [3328, 2112, 3392, 3904, 3328, 3264] | 58.8 |
| M27 | 5 | 960 | [8, 2, 8, 8, 4] | [4032, 3008, 3840, 3904, 3968] | 59.1 |
| M28 | 5 | 960 | [2, 4, 2, 2, 8] | [3968, 3968, 3840, 4032, 3904] | 60.9 |
| M29 | 6 | 896 | [2, 2, 4, 4, 2, 2] | [3840, 3968, 3840, 3328, 3904, 3904] | 65.0 |
| M30 | 6 | 960 | [4, 8, 8, 4, 8, 4] | [3072, 3584, 3392, 3840, 3328, 3712] | 67.9 |
| M31 | 6 | 960 | [4, 8, 8, 8, 4, 4] | [3840, 3584, 3392, 3328, 3968, 3776] | 69.7 |
| M32 | 6 | 960 | [4, 8, 8, 8, 8, 2] | [3840, 3840, 3392, 3840, 3328, 3712] | 69.9 |
| M33 | 6 | 960 | [4, 2, 2, 4, 2, 8] | [3840, 3008, 3840, 3904, 4032, 3392] | 70.0 |
| M34 | 6 | 960 | [2, 4, 8, 8, 4, 2] | [3840, 3968, 3840, 3328, 4032, 3776] | 71.5 |
| M35 | 7 | 960 | [4, 8, 8, 8, 8, 2, 8] | [3840, 3968, 3840, 3328, 3968, 3328, 4032] | 82.8 |
| M36 | 8 | 960 | [4, 2, 8, 8, 8, 4, 8, 8] | [3840, 3968, 3840, 3328, 3072, 3328, 4032, 3072] | 91.6 |
| M37 | 10 | 896 | [8, 4, 8, 8, 8, 2, 8, 2, 4, 8] | [4032, 3008, 3840, 2560, 3904, 3904, 3072, 3264, 2368, 2496] | 98.4 |
| M38 | 12 | 832 | [2, 4, 8, 8, 8, 8, 8, 8, 8, 8, 4, 2] | [3840, 2816, 2112, 3584, 3648, 2432, 2304, 3008, 2880, 1664, 2432, 3776] | 99.0 |
| M39 | 9 | 960 | [8, 8, 8, 4, 4, 8, 8, 4, 2] | [2752, 3456, 2880, 3904, 2752, 3904, 4032, 3264, 3136] | 99.3 |
| M40 | 10 | 896 | [8, 8, 8, 2, 8, 2, 2, 2, 8, 2] | [3840, 3072, 3840, 2560, 3648, 3328, 3840, 3008, 2880, 3328] | 100.0 |

(TITAN Xp)

Table 6: Detailed architectural parameters for all models in Figure 8 with GPT-2 backbone.

| | $n_{layer}$ | $d_{model}$ | $n_{head}$ | $d_{inner}$ | DecoderParams (M) |
|---|---|---|---|---|---|
| baseline | ∈[1,16] | 1024 | 12 | 3072 | - |
| M1 | 3 | 256 | [2, 2, 2] | [3072, 3776, 3904] | 6.3 |
| M2 | 2 | 448 | [2, 2] | [3456, 3776] | 8.1 |
| M3 | 2 | 448 | [2, 4] | [4032, 3904] | 8.7 |
| M4 | 3 | 384 | [2, 2, 2] | [3072, 2176, 4032] | 8.9 |
| M5 | 2 | 576 | [2, 2] | [3456, 3584] | 10.8 |
| M6 | 4 | 448 | [2, 2, 2, 2] | [4032, 3904, 1920, 3072] | 14.8 |
| M7 | 4 | 512 | [2, 2, 4, 2] | [3904, 3136, 1280, 2624] | 15.4 |
| M8 | 2 | 832 | [8, 2] | [3456, 3584] | 17.3 |
| M9 | 2 | 960 | [2, 8] | [3456, 3648] | 21.0 |
| M10 | 2 | 960 | [2, 2] | [3968, 3584] | 21.9 |
| M11 | 5 | 640 | [2, 2, 2, 2, 2] | [4032, 2560, 2176, 2304, 3136] | 26.4 |
| M12 | 3 | 832 | [2, 8, 4] | [3840, 3840, 3776] | 27.4 |
| M13 | 5 | 704 | [2, 2, 2, 4, 4] | [2368, 3648, 1856, 3712, 3200] | 30.8 |
| M14 | 3 | 960 | [2, 2, 2] | [3584, 3648, 4032] | 32.7 |
| M15 | 3 | 960 | [2, 2, 2] | [3904, 3520, 4032] | 33.1 |
| M16 | 6 | 640 | [2, 2, 2, 2, 2, 2] | [2624, 2560, 2880, 3776, 3648, 3840] | 34.6 |
| M17 | 4 | 896 | [2, 2, 4, 2] | [4032, 3712, 3328, 3072] | 38.2 |
| M18 | 5 | 832 | [2, 2, 2, 4, 4] | [3392, 3648, 2880, 3712, 3200] | 41.9 |
| M19 | 4 | 960 | [2, 2, 4, 2] | [3904, 3136, 3328, 3776] | 42.0 |
| M20 | 4 | 960 | [8, 8, 2, 4] | [3904, 3712, 4032, 3776] | 44.4 |
| M21 | 6 | 832 | [2, 2, 4, 2, 2, 2] | [3904, 3456, 4032, 1792, 3072, 2496] | 47.9 |
| M22 | 5 | 896 | [4, 2, 2, 2, 4] | [3968, 3200, 3840, 3328, 3648] | 48.3 |
| M23 | 5 | 960 | [2, 2, 2, 2, 2] | [3904, 3264, 3328, 3776, 3392] | 52.4 |
| M24 | 5 | 960 | [2, 2, 4, 2, 2] | [3584, 3456, 3776, 2944, 4032] | 52.7 |
| M25 | 5 | 960 | [2, 8, 2, 4, 2] | [3904, 3648, 4032, 3776, 3968] | 55.6 |
| M26 | 6 | 960 | [8, 8, 2, 2, 2, 2] | [3904, 2560, 2880, 3776, 2240, 3840] | 59.1 |
| M27 | 6 | 960 | [2, 2, 2, 4, 2, 2] | [2496, 3456, 3328, 3904, 3968, 2944] | 60.8 |
| M28 | 6 | 960 | [4, 2, 4, 4, 2, 8] | [4032, 3456, 3328, 3776, 4032, 2752] | 63.2 |
| M29 | 6 | 960 | [2, 2, 2, 4, 4, 4] | [3968, 3648, 3840, 3776, 3584, 2624] | 63.4 |
| M30 | 7 | 960 | [2, 2, 2, 4, 2, 4, 2] | [3904, 2368, 4032, 3008, 3520, 2944, 2496] | 68.7 |
| M31 | 7 | 960 | [2, 2, 4, 2, 2, 2, 4] | [3072, 3648, 3520, 3584, 3136, 1984, 3584] | 69.1 |
| M32 | 7 | 960 | [4, 2, 2, 2, 8, 2, 2] | [3712, 3648, 3584, 3520, 2752, 3008, 3392] | 71.2 |
| M33 | 8 | 960 | [2, 4, 4, 2, 2, 2, 2, 2] | [3904, 2816, 3072, 1920, 3328, 3456, 2304, 2368] | 74.1 |
| M34 | 8 | 960 | [2, 2, 2, 4, 2, 2, 8, 2] | [3520, 2368, 4032, 1792, 3200, 3776, 3200, 3648] | 78.6 |
| M35 | 8 | 960 | [4, 2, 4, 4, 8, 8, 4, 2] | [3520, 3712, 3328, 3776, 3200, 2752, 3200, 2112] | 78.7 |
| M36 | 8 | 960 | [8, 4, 2, 8, 2, 2, 2, 2] | [3520, 3840, 3328, 3776, 3200, 3776, 3968, 3648] | 85.4 |
| M37 | 10 | 960 | [2, 8, 2, 4, 2, 2, 4, 2, 8, 8] | [3648, 2560, 3776, 1792, 3968, 2752, 3200, 2368, 4032, 2368] | 95.5 |
| M38 | 10 | 960 | [2, 4, 2, 2, 4, 2, 4, 2, 4, 8] | [3840, 2240, 3328, 3776, 3648, 3200, 2944, 2368, 3968, 2880] | 98.8 |
| M39 | 10 | 960 | [2, 4, 2, 2, 2, 2, 4, 2, 4, 8] | [3840, 2240, 3328, 3776, 3200, 3200, 3968, 2368, 3968, 2816] | 99.8 |

(TITAN Xp)

Table 7: Detailed architectural parameters for all models in Figure 8 with GPT-2 backbone.

| | | $n_{layer}$ | $d_{model}$ | $n_{head}$ | $d_{inner}$ | DecoderParams (M) |
|---|---|---|---|---|---|---|
| | baseline | $\in[1,16]$ | 1024 | 12 | 3072 | - |
| ARM | M1 | 2 | 512 | [2, 2] | [1920, 1920] | 6.0 |
| | M2 | 3 | 320 | [8, 2, 4] | [1920, 1920, 3712] | 6.1 |
| | M3 | 2 | 576 | [2, 2] | [1344, 3200] | 7.9 |
| | M4 | 3 | 384 | [2, 8, 2] | [3840, 2368, 3328] | 9.1 |
| | M5 | 5 | 384 | [4, 4, 2, 4, 4] | [2880, 1920, 960, 2496, 1280] | 10.3 |
| | M6 | 2 | 768 | [2, 2] | [1600, 2240] | 10.6 |
| | M7 | 5 | 320 | [4, 2, 2, 4, 2] | [1344, 2240, 3776, 3008, 3648] | 11.0 |
| | M8 | 3 | 768 | [2, 2, 4] | [1856, 1792, 1920] | 15.7 |
| | M9 | 3 | 704 | [2, 2, 2] | [3136, 2112, 1920] | 16.1 |
| | M10 | 2 | 960 | [4, 2] | [3584, 2304] | 18.7 |
| | M11 | 6 | 448 | [4, 4, 2, 2, 4, 2] | [3072, 2112, 4032, 2688, 1600, 3072] | 19.7 |
| | M12 | 3 | 960 | [4, 4, 2] | [2368, 2560, 2048] | 24.5 |
| | M13 | 4 | 704 | [4, 8, 4, 2] | [3008, 3776, 2560, 3648] | 26.3 |
| | M14 | 5 | 704 | [4, 2, 4, 2, 8] | [3584, 3136, 3776, 3072, 1856] | 31.7 |
| | M15 | 3 | 960 | [2, 2, 2] | [3392, 3648, 3840] | 32.0 |
| | M16 | 4 | 960 | [4, 2, 8, 2] | [2048, 3328, 1984, 1856] | 32.5 |
| | M17 | 7 | 704 | [2, 4, 4, 4, 8, 2, 2] | [3008, 2560, 1920, 1856, 2112, 1728, 3136] | 36.9 |
| | M18 | 4 | 960 | [2, 2, 4, 8] | [3392, 3456, 2432, 2304] | 37.0 |
| | M19 | 5 | 832 | [4, 4, 4, 4, 4] | [3840, 1920, 4032, 3072, 3968] | 41.9 |
| | M20 | 5 | 960 | [8, 4, 2, 2, 4] | [2560, 2048, 3648, 1728, 2304] | 42.1 |
| | M21 | 5 | 960 | [4, 4, 2, 2, 2] | [3072, 2240, 1984, 2176, 3520] | 43.4 |
| | M22 | 5 | 960 | [2, 4, 4, 4, 2] | [2496, 3648, 3328, 3392, 2112] | 47.2 |
| | M23 | 6 | 832 | [4, 2, 4, 4, 2, 4] | [2496, 3200, 1664, 3904, 3520, 3840] | 47.7 |
| | M24 | 6 | 960 | [8, 2, 2, 2, 8, 4] | [2304, 3328, 3456, 1856, 1792, 2112] | 50.7 |
| | M25 | 5 | 960 | [4, 8, 2, 4, 4] | [3264, 2688, 4032, 3968, 3712] | 52.4 |
| | M26 | 6 | 960 | [2, 4, 4, 2, 2, 2] | [3008, 2624, 4032, 2688, 3520, 2624] | 57.7 |
| | M27 | 6 | 960 | [2, 4, 4, 2, 8, 2] | [2304, 3648, 3328, 3648, 3904, 1728] | 57.8 |
| | M28 | 6 | 960 | [4, 4, 2, 4, 2, 2] | [3072, 2368, 4032, 4032, 3776, 3264] | 61.6 |
| | M29 | 7 | 960 | [2, 2, 2, 8, 4, 8, 4] | [3008, 2304, 1920, 1984, 3520, 2816, 3712] | 62.9 |
| | M30 | 7 | 960 | [2, 4, 4, 4, 4, 2, 2] | [3200, 4032, 2048, 2624, 2112, 2752, 2880] | 63.6 |
| | M31 | 7 | 960 | [2, 4, 4, 4, 2, 4] | [3584, 3648, 3328, 3392, 3200, 1984, 3200] | 68.8 |
| | M32 | 7 | 960 | [2, 4, 8, 8, 2, 2, 8] | [3008, 3648, 3584, 3648, 3008, 1728, 3712] | 68.8 |
| | M33 | 7 | 960 | [4, 4, 2, 4, 4, 8, 4] | [3584, 3840, 3328, 3392, 3136, 2944, 2496] | 69.5 |
| | M34 | 8 | 960 | [8, 2, 2, 8, 2, 8, 2] | [3008, 3648, 1792, 1984, 3008, 2816, 3712, 3520] | 74.7 |
| | M35 | 8 | 960 | [2, 2, 2, 2, 8, 4, 4, 2] | [3008, 2304, 1792, 3008, 3520, 2880, 3712, 3456] | 75.1 |
| | M36 | 8 | 960 | [2, 2, 2, 2, 2, 4, 8] | [3008, 1792, 3840, 3392, 3520, 3136, 3712, 3520] | 79.4 |
| | M37 | 9 | 960 | [2, 2, 4, 4, 8, 8, 4, 2, 4] | [1664, 1792, 2240, 3904, 3648, 3264, 2176, 3712, 1856] | 79.9 |
| | M38 | 11 | 832 | [8, 4, 2, 4, 4, 2, 8, 4, 4, 8, 8] | [3072, 2368, 4032, 3968, 1664, 3968, 2176, 2624, 3840, 2176, 2112] | 83.8 |
| | M39 | 9 | 960 | [4, 2, 4, 8, 2, 2, 4, 2, 4] | [2496, 3648, 3328, 3392, 3648, 1728, 2880, 3520, 2368] | 85.1 |
| | M40 | 9 | 960 | [4, 2, 4, 8, 4, 2, 4, 2, 4] | [3072, 2816, 4032, 2560, 3648, 1728, 3840, 3264, 3456] | 87.8 |
| | M41 | 10 | 960 | [8, 2, 4, 4, 2, 2, 4, 8, 2, 4] | [3648, 1792, 2432, 1856, 3392, 2304, 3776, 2944, 3136, 3904] | 93.0 |
| | M42 | 10 | 960 | [8, 2, 2, 4, 2, 2, 2, 4, 2, 2] | [3264, 2048, 3520, 3904, 3840, 3840, 2624, 3072, 3776, 2304] | 98.8 |
| | M43 | 12 | 896 | [4, 4, 4, 2, 4, 2, 4, 8, 8, 2, 4, 2] | [2048, 3136, 4032, 1792, 3584, 1728, 3136, 3008, 2560, 3200, 3648, 1728] | 98.9 |
| | M44 | 10 | 960 | [4, 2, 8, 4, 2, 8, 4, 4, 4, 2] | [3584, 3968, 3328, 3904, 2368, 2112, 3904, 3520, 3328, 2688] | 99.8 |
| | M45 | 10 | 960 | [8, 2, 4, 4, 4, 4, 4, 2, 2, 8] | [2688, 3200, 3840, 3392, 3520, 3136, 3392, 3520, 2880, 3200] | 99.9 |
| Corei7 | M1 | 2 | 384 | [2, 2] | [3840, 2432] | 6.0 |
| | M2 | 3 | 320 | [2, 2, 2] | [2176, 3072, 2496] | 6.2 |
| | M3 | 2 | 512 | [2, 2] | [1408, 2624] | 6.2 |
| | M4 | 3 | 384 | [2, 2, 2] | [3264, 3456, 3584] | 9.7 |
| | M5 | 2 | 576 | [2, 2] | [3136, 3648] | 10.5 |
| | M6 | 3 | 448 | [2, 2, 2] | [4032, 3648, 4032] | 12.9 |
| | M7 | 4 | 448 | [2, 2, 4, 4] | [3072, 3648, 4032, 1792] | 14.5 |
| | M8 | 2 | 768 | [2, 2] | [3968, 3328] | 15.9 |
| | M9 | 4 | 576 | [2, 2, 2, 2] | [3072, 2752, 3456, 3136] | 19.6 |
| | M10 | 2 | 960 | [2, 2] | [3840, 3264] | 21.0 |
| | M11 | 4 | 640 | [2, 2, 2, 2] | [2176, 3648, 3584, 1920] | 21.1 |
| | M12 | 3 | 960 | [2, 2, 2] | [2176, 3264, 2432] | 26.2 |
| | M13 | 4 | 768 | [2, 2, 2, 2] | [3584, 2112, 3392, 1920] | 26.4 |
| | M14 | 4 | 768 | [2, 2, 2, 2] | [3584, 2560, 3776, 1536] | 27.1 |
| | M15 | 4 | 832 | [2, 2, 2, 2] | [3904, 1984, 3392, 3136] | 31.8 |
| | M16 | 3 | 960 | [2, 2, 2] | [3968, 4032, 2880] | 32.0 |
| | M17 | 5 | 768 | [2, 2, 4, 2, 2] | [3648, 3072, 3392, 1984, 2944] | 34.9 |
| | M18 | 4 | 960 | [2, 2, 2, 2] | [3136, 1984, 3392, 2944] | 36.8 |
| | M19 | 4 | 960 | [2, 2, 2, 4] | [3968, 3456, 3584, 3136] | 42.0 |
| | M20 | 6 | 768 | [4, 2, 2, 4, 2, 4] | [3584, 2112, 3456, 3136, 3840, 2560] | 42.9 |
| | M21 | 7 | 768 | [2, 4, 2, 4, 4, 4, 2] | [2624, 1984, 2496, 3968, 2880, 2112, 4032] | 47.5 |
| | M22 | 5 | 960 | [2, 2, 4, 2, 4] | [2176, 3264, 3392, 3008, 3328] | 47.6 |
| | M23 | 6 | 960 | [4, 4, 2, 4, 2, 2] | [2048, 2624, 3520, 1984, 2880, 2624] | 52.3 |
| | M24 | 6 | 960 | [2, 4, 4, 4, 2, 2] | [1792, 3456, 2752, 2240, 1664, 3840] | 52.4 |
| | M25 | 6 | 960 | [4, 2, 2, 2, 4, 4] | [2176, 1664, 3648, 3136, 3968, 3904] | 57.7 |
| | M26 | 7 | 960 | [2, 2, 4, 4, 2, 2, 8] | [2816, 1792, 3968, 1728, 1664, 3328, 2944] | 60.9 |
| | M27 | 7 | 896 | [2, 2, 4, 2, 2, 2, 2] | [3904, 3264, 3328, 3968, 1728, 2624, 4032] | 63.5 |
| | M28 | 7 | 960 | [4, 2, 4, 2, 2, 2, 2] | [3584, 2560, 1792, 1920, 3968, 2112, 3968] | 64.1 |
| | M29 | 8 | 960 | [2, 2, 2, 4, 2, 2, 2, 4] | [3328, 2432, 2624, 2752, 1664, 2240, 2304, 2816] | 68.3 |
| | M30 | 7 | 960 | [4, 2, 4, 2, 2, 2, 2] | [3904, 2304, 2368, 3584, 3264, 2880, 3904] | 68.5 |
| | M31 | 8 | 960 | [4, 2, 4, 2, 4, 2, 4, 4] | [2560, 3648, 2624, 2112, 3328, 2112, 1792, 3328] | 70.9 |
| | M32 | 8 | 960 | [4, 4, 4, 2, 2, 4, 2, 4] | [2560, 2304, 2624, 4032, 2688, 2624, 3840, 2816] | 74.7 |
| | M33 | 9 | 960 | [2, 4, 2, 4, 2, 4, 2, 2, 4] | [3072, 3264, 2944, 1984, 2880, 3520, 2112, 2624, 1728] | 79.6 |
| | M34 | 10 | 896 | [2, 2, 4, 2, 2, 2, 2, 4, 2] | [2816, 3264, 3584, 1792, 3136, 3584, 2240, 2240, 1920, 2752] | 81.2 |
| | M35 | 9 | 960 | [8, 2, 2, 2, 4, 4, 2, 4, 4] | [3904, 3648, 2432, 3136, 3264, 2816, 2240, 3072, 3840] | 87.7 |
| | M36 | 10 | 960 | [4, 2, 2, 4, 4, 2, 4, 4, 4, 2] | [2176, 3264, 2752, 3136, 3968, 3520, 3776, 3328, 1728, 2496] | 94.9 |
| | M37 | 10 | 960 | [4, 2, 4, 2, 2, 2, 2, 4, 2, 2] | [3904, 2112, 2496, 3968, 3968, 2624, 3904, 2304, 3200, 3840] | 99.0 |
| | M38 | 11 | 960 | [4, 2, 2, 4, 2, 4, 2, 2, 4, 4, 4] | [2176, 4032, 3264, 3840, 2688, 1984, 1728, 2944, 1920, 2368, 3840] | 99.8 |