# OpenReview forum: "LiteTransformerSearch: Training-free Neural Architecture Search for Efficient Language Models"
_NeurIPS.cc/2022/Conference — NeurIPS 2022 Accept_

### Official Review · Reviewer_KSpx · 2022-07-04

**Rating:** 5
**Confidence:** 5
**Soundness:** 3 good
**Presentation:** 3 good
**Contribution:** 2 fair

**Summary:**

This paper adapts the number of decoder parameters as the search proxy to accelerate the design loop for decoder-only language models. The authors verified the assumption using 2800 trained data points  and based on the assumption they show 1.6-2.5x faster runtime speed and 1.3-2.0x lower peak memory on various hardware.

**Questions:**

In Figure 3, WikiText-103 is a relative small dataset compared to standard GPT-2 level model trainign (e.g., WikiText of billions of tokens). If the dataset size becomes larger, will the precision of #num_of_parameters proxy become better or worse?

In Figure 8, the authors show PPL on evaluation. However, in Algorithm 1 there is no description about how the model is trained. Is every model trained separately, or there is parameter sharing techiniques?

**Limitations:**

Overall this is a good paper with clear method and solid experiment results. If the authors can address the aformentioned weakness above, I would like to improve the score.

**Strengths And Weaknesses:**

Strength

- The paper tackles on an important problem.
- The problem formulation is clear and writing is well presented.
- Evaluation in Figure 3 shows the correlation between #num parameters and final model performance.
- The pareto-frontier exploration demonstrates significant improvements compared with naive transformer scaling.


Weakness

- The biggest assumption highly relies on decoder-only architecture. L377: Our focus is entirely on autoregressive, decoder-only transformer. This is not a general solution to explore NAS on tasks like NMT.
- Only compared with scaled transformers, which is not considered as strong baseline. Since this is NAS work, the latency and accuracy from models derived with AutoML techniques should be discussed.
- Free proxy does not mean free evaluation — the generated architecture still need to perform full training to obtain the model weights. Then even the search cost is reduced, the total cost will not be saved much compared to once-for-all studies like [1, 2].

1. Hat: Hardware-aware transformers for efficient natural language processing.
2. Prune Once for All: Sparse Pre-Trained Language Models

---

> ### Author Response · Authors · 2022-08-02
> **Response to reviewer KSpx (part 1)**
>
> We thank the reviewer for their kind words about the quality of the paper and our experiments. Below we address the weaknesses and questions in detail. We hope we have addressed the raised questions to your satisfaction. Please let us know if there are any further questions that we can help address during the interactive rebuttal period.
>
> 1.  **Weakness 1 (focus on autoregressive Transformers):** we have preliminary results on the adoption of our training-free, parameter-count-based proxies for other tasks (please see our answer number 2 below) but careful consideration requires a large number of experiments which will warrant an entire paper of its own. Due to their fundamental differences, studying each class of Transformer models is a challenge that requires extensive GPU hours of training. As a result, the existing papers on NAS for Transformers, only study one target application. As an example, HAT [B] focuses only on NMT, i.e., NAS on Transformers with encoder-decoder blocks, and does not provide results on autoregressive models. Alternatively, Primer [C] only focuses on autoregressive Transformers with decoder blocks. Existing work on scaling laws for Transformers also only study one task, e.g., [D,E] for autoregressive Transformers, [F] for NMT, [G] for transfer learning on autoregressive Transformers. Due to the empirical challenge and in line with prior works, we also limit our comprehensive studies to autoregressive models.
>
> 2.  **Weakness 1 (generalizability of the solution):** Scaling laws for various classes of Transformers have recently gained traction in recent papers (see above for details) [D,E,F,G]. Our paper is the first work that connects NAS with such scaling laws and shows that they can be leveraged to develop training-free proxies (or multi-fidelity approaches) for architecture search. As such, it can be the starting point for future research that apply the same findings to other classes of Transformers and NLP applications. Specifically, we have preliminary results which show a high correlation of **0.96** exists between encoder parameter count and performance (equivalently loss) in randomly sampled Bert-style Transformers (encoder-only models trained with a mixture of masked language modeling and next sentence prediction). Additionally, we have conducted new experiments on the open-sourced NMT models in [I] which show a high correlation of **0.87** exists between non-embedding parameter count (summation of parameters in encoder and decoder) and bleu score on ja-en translation task. Interestingly on so-en and sw-en translation tasks, the non-embedding parameter count is inversely correlated with bleu score with spearman rank correlation of **0.85** and **0.81**, which shows smaller NMT models are better suited when training data is limited.
> We have added the above results on the BERT backbone and NMT domain in a new Appendix Section I in the revised manuscript. These preliminary studies validate that the proposed training-free proxy can be used as a strong baseline for NAS in other domains but certainly, more in-depth analyses are required which is out-of-scope of this particular work.
>
> 3.  **Weakness 2 (comparison baseline):** Unfortunately to the best of our knowledge, there are no prior works that apply AutoML/NAS to autoregressive Transformers which matches our setting. Therefore, the only viable baseline for comparison is scaled GPT2 or Transformer-XL models. Using scaled backbone Transformers as the baseline is done in prior works [B,C], when no other NAS baselines are available. Please note that the models extracted by HAT are not comparable with LTS since they are encoder-decoder architectures and the target application is NMT, therefore it is not possible to compare them with our decoder-only models in terms of perplexity and/or latency. The only existing work on AutoML/NAS for decoder-only Transformers is Primer [C], however, their search objective is to find models that train faster, not models that have lower latency/memory during inference and they use a program-synthesis style search space that is entirely different from the more common transformer backbone which we consider. Therefore, Primer targets an entirely different pareto of models which can not be directly compared with our models in any aspect. Such issues are endemic in the entire NAS field which has motivated the appearance of NAS benchmarks for easy comparison [J,K]. Since there are no benchmarks available for autoregressive Transformers to date, we hope that open-sourcing our trained models can alleviate this issue and provide a strong baseline for future works.

---

> > ### Author Response · Authors · 2022-08-02
> > **Response to reviewer KSpx (part 2)**
> >
> > 4.  **Weakness 3 (cost of training):** The reviewer is absolutely correct that the model needs to be trained after the search is completed. After the pareto-frontier models are found using training-free search, the users typically pick one model that satisfies their needs (in terms of latency/memory/perplexity) and train it. In this scenario, the cost of training this model will be much less than training a supernet or once-for-all model (OFA) since the supergraph is the size of the biggest model in the search space. As an example, if the user has a latency budget of 10 ms on a Titan Xp GPU then in Figure 8 one simply drops a vertical line from 10 ms on the x-axis up and picks the nearest model to the left of this line on the Pareto curve and trains it. Alternatively, if the user requires the lowest latency model that has a validation perplexity of no more than 40, then one drops a horizontal line from 40 ppl and picks the model closest to the intersecting point with the Pareto-frontier to train. If the user is interested in training all the models on the pareto-frontier, they can create a supernet using the largest configuration on the pareto and train it with the once-for-all methodology [B,H]. Although we have not done this in the paper, the OFA training is applicable to our pareto-frontier models and is orthogonal to the benefits of our efficient search.
> >
> > 5.  **Question 1 (effect of dataset size):** In addition to WikiText-103, we perform our evaluations on the one billion word dataset (LM1B in Figure 3 and 8), which is among the largest publicly available language modeling datasets. As outlined in [A], the number of training tokens in this dataset are adequately large for training the range of model sizes in this paper. As such, increasing the dataset size beyond LM1B should not affect the final performance of our models, i.e., the effectiveness of the parameter count proxy remains unchanged. Please see items number 1 and 2 in our response to reviewer Zj8C for more details on the maximal dataset size per model.
> >
> > 6.  **Question 2 (Figure 8 training):** We have trained each architecture in this Figure separately from scratch. In many scenarios, the user only needs to train one model from the Pareto-frontier, which is selected based on their needs for perplexity, latency, and memory. However, if the users are interested in multiple models, they can either train all models separately or create a supergraph enclosing all such configurations and train them all simultaneously using weight sharing as in [B,H]. We have added this clarification to the revised paper in Appendix B.
> >
> > [A] [Training Compute-Optimal Large Language Models](https://arxiv.org/pdf/2203.15556v1.pdf), 2022.
> >
> > [B] [HAT: Hardware-aware transformers for efficient natural language processing](https://arxiv.org/pdf/2005.14187.pdf), ACL 2020.
> >
> > [C] [Primer: Searching for Efficient Transformers for Language Modeling](https://openreview.net/pdf?id=bzpkxS_JVsI), Neurips 2021.
> >
> > [D] [Scaling laws for autoregressive generative modeling](https://arxiv.org/pdf/2010.14701.pdf), 2020.
> >
> > [E] [Scaling laws for neural language models](https://arxiv.org/pdf/2001.08361.pdf), 2020.
> >
> > [F] [Scaling laws for neural machine translation](https://openreview.net/pdf?id=hR_SMu8cxCV), ICLR 2022.
> >
> > [G] [Scaling Laws for Transfer](https://arxiv.org/pdf/2102.01293.pdf), 2022.
> >
> > [H] [Prune Once for All: Sparse Pre-Trained Language Models](https://arxiv.org/pdf/2111.05754.pdf), 2021.
> >
> > [I] https://github.com/Este1le/hpo_nmt/tree/master/automl2022
> >
> > [J] [Nas-bench-201: Extending the scope of reproducible neural architecture search](https://arxiv.org/pdf/2001.00326.pdf), ICLR 2020.
> >
> > [K] [Surrogate NAS benchmarks: Going beyond the limited search spaces of tabular NAS benchmarks](https://openreview.net/pdf?id=OnpFa95RVqs), ICLR 2022.

---

> > > ### Comment · Reviewer_KSpx · 2022-08-07
> > > **Thanks for your response and few more questions**
> > >
> > > After carefully reading the view, some of my previous concerns have been addressed. The figure in revised appendix answers my question about the relation between params and performance.
> > >
> > > One remaning question I had is about the evalution process -- After a model is obtained, the model requires to be trained from scratch and such cost is not negligible. The authors say "the OFA training is applicable to our pareto-frontier models and is orthogonal to the benefits of our efficient search". But with OFA training, the performance can directly evaluated with subtracted weights and why we need extra proxy to search?

---

> > > > ### Author Response · Authors · 2022-08-07
> > > > **Answer to your additional question**
> > > >
> > > > We thank the reviewer for reading our response and are glad we were able to address their questions. Below we answer the new question:
> > > >
> > > > It is true that once the models along the pareto-frontier have been obtained, they indeed need to be trained from scratch and this is not a negligible cost. What we meant by the OFA comment is merely using their concept of “weight-sharing” to fuse a few of the pareto-frontier architectures together and train them simultaneously. E.g., if the user wants to train say two particular models from the pareto-frontier and they have 3 and 5 layers respectively, the user can fuse them into **one** 5-layer model and train both at the same time using weight sharing. The cost of training this supergraph remains roughly the same as training a 5-layer model. But note that this simple trick to amortize training cost does not mean that the trained 5-layer model can be used to rank all the sampled architectures during search (which is the more widely known use of OFA/weight-sharing [A,B,C]). Since the 5-layer model is created by fusing only a few pareto-frontier models and only those few sub-architectures are sampled during the training. This is mainly to save the cost of training the pareto-frontier models individually. In practice, the user will only train the single model on the pareto-frontier that best meets their constraints (e.g. latency <20 ms) so the cost of the entire procedure should be gpu-free fast search followed by training from scratch the model with the best trade-offs for user application on gpu. We will incorporate this discussion in the paper as well for more clarity.
> > > >
> > > > If the reviewer’s concerns have been satisfied will they kindly reconsider their score? Thanks in advance!
> > > >
> > > > [A] [Evaluating Efficient Performance Estimators of Neural Architectures](https://openreview.net/pdf?id=Esd7tGH3Spl), Neurips 2021.
> > > >
> > > > [B] [Bossnas: Exploring hybrid cnn-transformers with block-wisely self-supervised neural architecture search](https://openaccess.thecvf.com/content/ICCV2021/papers/Li_BossNAS_Exploring_Hybrid_CNN-Transformers_With_Block-Wisely_Self-Supervised_Neural_Architecture_Search_ICCV_2021_paper.pdf), ICCV 2021.
> > > >
> > > > [C] [HAT: Hardware-aware transformers for efficient natural language processing](https://arxiv.org/pdf/2005.14187.pdf), ACL 2020.

---

> > > > > ### Comment · Reviewer_KSpx · 2022-08-08
> > > > > **Response**
> > > > >
> > > > > Thanks for the detailed update. I would recommend authors incooperate discussed points into main paper in final version. Given the solid justfication, I would like to raise my score from *borderline accept*.

---

> > > > > > ### Author Response · Authors · 2022-08-08
> > > > > > **Thanks for your feedback**
> > > > > >
> > > > > > Thank you so much for your response and for engaging with us during the discussion period. To comply with the revision page limit we had to put all new discussions in the Appendix for now. But per your suggestion, we will move all the discussions into the main paper for the final version by using the extra one page for camera-ready.

---

### Official Review · Reviewer_zV4x · 2022-07-09

**Rating:** 6
**Confidence:** 3
**Soundness:** 3 good
**Presentation:** 3 good
**Contribution:** 2 fair

**Summary:**

This paper proposes to use the number of decoder parameters as a training-free metric for autoregressive transformer architecture search for language models. The paper then shows that the metric correlates strongly with the final perplexity of the model over the entire model sizes and topologies. Because of the training-free nature of the proxy proposed, the authors show that using otherwise standard search algorithms such as evolutionary search, it is possible to optimise for architectures that trade off perplexity, memory and latency on devices with modest computational power directly, and it is possible to recover much better Pareto frontier using the method proposed compared to the naive scaling of baseline architectures.


**Questions:**

Please see Strengths And Weaknesses

**Limitations:**

The authors have adequately discussed the limitations of their work both in Conclusion and in appendix. The authors should also be given credit for interleaving several broader impact discussions in the paper (rather than as a standalone section), such as the estimation of both GPU hours and $CO_2$ emission statistics in Table 1.

**Strengths And Weaknesses:**

Overall I believe this is a sound paper with potential for real impact, and the writing and presentation are clear and good throughout. I have outlined my detailed comments on strengths and weaknesses below.  Note that I have combined this section with my questions (next section) as many of them are related, and given the time constraint for the rebuttal, I do not expect the authors to respond thoroughly to all of my questions and concerns especially where additional experiments are necessary; in these cases some qualitative response would suffice.


## Strengths

The algorithm proposed is simple and effective in the search spaces investigated by the authors: the number of decoder parameters is very cheap to obtain, and is arguably cheaper than many of the previous zero- or low-cost proxies proposed in the NAS literature (also considered by the authors in Fig 5). The ability to directly search for better architectures on target devices with computational constraints as a result of the lowered computational cost can lead to very tangible benefits, so this paper has potential to be very impactful.


## Weaknesses & questions

1. A concern I have is that the fact that performance (perplexity in the NLP context) is correlated with some measure of #params is highly expected in general and is also demonstrated to be the case in NAS in particular (see the results of the size search space of NATS-Bench, an extension of NAS-Bench-201 [1]). In many cases, some measure of #params has been used as a search objective alongside performance [2] (not in this case, as the authors used mem usage and latency); this shows that the correlation between that and performance is assumed. While the paper does show some nuance by demonstrating the superiority of #decode params over the total #params (Fig 6), I am still a bit unconvinced by the fact that most of the analysis (Sec 4.1 - 4.3) use the absolute performance as the outcome variable, even though ultimately we are interested in the *trade-off* between performance and #params by finding a family the architectures that maximise performance at different part of the Pareto frontier instead of just maximising performance in Sec 4.4. It is worth noting that the existing zero-cost proxies the authors compared like synflow and jacob_norm were largely originally evaluated in cell-based search spaces where we often only aim to find a single architecture with its accuracy optimised [3,4], and that’s why it is more justified to use the absolute performance as the dependent variable in that case.

      Given this goal of finding Pareto-optimal family of architectures, I feel that analyses should instead pivot on the predictive ability of the # decoder params after the effect of other confounding variables are removed: for example, how well does #decoder param predicts perplexity on architectures with similar mem usage and/or latency (using a metric such as the partial correlation)? I feel this is a more direct test of how well the proposed proxy can truly help discover more Pareto-optimal architectures. While the authors did discuss these briefly in Appendix C, I believe these analyses are rather important and should be expanded: for example, 1) the paper considers both latency and mem_usage as optimization objectives, but App C only considers latency; 2) similar to how NAS predictor can often predict well over the entire search space but break down in high-accuracy regions (which are also what we care about), the proxy proposed might also suffer degradations nearer to the Pareto frontier and it would be interesting to investigate whether that is the case and 3) the results in App C do show that the proxy quality is worse at larger models, and it could be interesting to explain/investigate why it might be the case and discuss potential remedies.

    On a related note, I feel it would also be good for the authors to include some explanations or intuitions on why do they think why # decoder params is a great proxy in this case. I feel that this could potentially be inspirational for future works that would like to develop on the idea but, for example, on a different search space or domain of application. It is also helpful in this particular case as this is the main innovation (the search techniques and other part of the algorithms are otherwise standard).

2. Experiments: Sec 4.4 is where the main experiments are demonstrated to show promise of the method. My main concern is that while the results are promising, the baselines seem a bit weak that the authors simply scale the base, homogenous Transformer and GPT models. I am wondering whether it is possible also to compare against a single or a family of NAS architectures from, for example, one or more of the prior works the authors cited in Related Works?

3. Compatibility/sensitivity with varying training hyperparameters: One thing also noted by the authors is that the best Transformers is often obtained by also tuning the training hyperparameters in addition to the architectural hyperparamaters (footnote 2). One interesting thing to investigate is whether the the suggested proxy still predict performance well when there are changes in hyperparameters.

4. Related work: there is also a recent work that proposes training-free transformer search proxy [5]. While I recognise that since this paper is only published at CVPR 2022 it is concurrent to the current submission (and my rating is independent of this paper), the authors are encouraged to include some qualitative discussions & comparisons w.r.t this work.

5. Train code: the authors have not provided code for the paper and mention they will open source the model upon acceptance. Would you please also clarify are you going to open-source the code? I think it would greatly enhance the reproducibility & benefit the broader community more if both code and models will be released.


## References

[1] Dong, X., Liu, L., Musial, K., & Gabrys, B. (2021). Nats-bench: Benchmarking nas algorithms for architecture topology and size. IEEE transactions on pattern analysis and machine intelligence.

[2] Chen, M., Peng, H., Fu, J., & Ling, H. (2021). Autoformer: Searching transformers for visual recognition. In Proceedings of the IEEE/CVF International Conference on Computer Vision (pp. 12270-12280).

[3] Abdelfattah, M. S., Mehrotra, A., Dudziak, Ł., & Lane, N. D. (2021). Zero-cost proxies for lightweight NAS. ICLR.

[4] White, C., Zela, A., Ru, R., Liu, Y., & Hutter, F. (2021). How powerful are performance predictors in neural architecture search?. Advances in Neural Information Processing Systems, 34, 28454-28469.

[5] Zhou, Q., Sheng, K., Zheng, X., Li, K., Sun, X., Tian, Y., ... & Ji, R. (2022). Training-free Transformer Architecture Search. In Proceedings of the IEEE/CVF Conference on Computer Vision and Pattern Recognition (pp. 10894-10903).

---

> ### Author Response · Authors · 2022-08-02
> **Response to reviewer zV4x (part 1)**
>
> We thank the reviewer for their insightful comments and encouraging words about the impact of our paper. Below we address all comments in detail. Please let us know if there are any further questions that we can address during the interactive rebuttal period.
>
>  1. **Weakness 1 (predictive ability of decoder params after removing the effect of other variables):** We test our proxy on a large number of random models, which cover various latencies, memories, and performances (Figures 3 and 9). This setting resembles the problem of finding the pareto-frontier, since it is not enough to only compare models with the same latency/memory, as the pareto-frontier is extracted by looking holistically at all neighboring and non-neighboring points in the multi-dimensional space which may have different latencies, memories, and perplexities. But we were curious to test the reviewer’s suggestion so we categorized all models in Fig. 9 into different latency bins, where the maximum difference inside each bin is 20 ms. We computed the rank correlation between decoder parameter count and validation perplexity within each bin. The average spearman correlation is 0.73, which is quite high given the small nuances in the models and their perplexities within each latency bin.
>
> -   **Weakness 1 (point 1):** The experiment in Appendix C fully trains all (>1000) models visited during search, therefore incuring a very high cost (both in terms of GPU hours and Co2 emission). We, therefore, left out the memory axis since it leads to an exponential increase in the number of models to train. But we are confident that a similar result would have held with memory as a 3rd axis.  Fig. 9 conveys that decoder parameter count can accurately represent perplexity in a multi-objective search. The addition of memory, or any arbitrary hardware metric, will not affect this finding, since our proxy is independent of the hardware metrics and the high correlation with perplexity is verified on numerous models. Specifically, in the pareto-frontier, latency and memory are not estimated but rather evaluated with real measurements. So the only objective that uses a proxy is validation perplexity, which we have vetted thoroughly in the paper and also jointly with latency in Fig. 9.
>
> -   **Weakness 1 (point 2):** Our experiment in Fig. 9 Appendix C aims to answer this question. Specifically, we wanted to validate whether the decoder parameter count proxy can be reliable to find the pareto-frontier, and how much distance does the proxy-based pareto-frontier have with the ground-truth. We therefore fully trained all models encountered during a complete search to plot the validation perplexity versus latency. We show that the models marked as pareto-forntier by the proxy are either on the true pareto-frontier, or have very small distance from it (0.6%). If the proxy had broken down near the pareto-frotnier the distance from the ground-truth pareto-frontier could be arbitrarily large. We would love the reviewer's input on whether they think this suffices for checking the ability of the proxy to find a good pareto-frontier. Is there a better alternative analysis they suggest?
>
> -   **Weakness 1 (point 3):** The slight downgrade in the performance of the proxy for larger models can be due to the plateau effect of perplexity, i.e., as models pass a certain size, the changes in the validation perplexity become very subtle when moving from one architecture to another. Therefore, it is harder to distinguish between models using a proxy metric as shown in Fig. 10 Appendix C. Nevertheless, the proxy allows us to find good models both in the small and large parameter regime as shown in the paretos of Fig. 8. This is also an insight noted in [C] which shows that even much smaller rank correlations can lead to a successful NAS.
>
> -   **Weakness 1 (providing Intuition):** Given that the nature of (linear and nonlinear) operations performed on the input remains the same for all models in a Transformer-based search-space, it is perhaps intuitive that a higher capacity model can better generalize to the underlying data distribution. We were inspired by the recent line of work in scaling laws for Transformers [D,E,F,G], which show that the performance can be modeled with simple power laws. Nevertheless, the connection between such scaling laws and NAS had not been explored before and the advances in the field of NAS have largely been made without consideration of such phenomenon. We have provided preliminary results in answer 2 to weakness 1 by reviewer KSpx that show scaling laws can be used as training-free or low-cost proxies for NAS in new domains, e.g., NMT, which is an interesting future work. We aim to influence the NAS field to consider training-free proxies not only as a strong baseline but also to use in multi-fidelity search [H] to rapidly guide the search to a strong part of the search-space where partial evaluation can be used to zoom-in further.

---

> > ### Author Response · Authors · 2022-08-02
> > **Response to reviewer zV4x (part 2)**
> >
> > 2.  **Weakness 2 (comparison baselines):** Unfortunately the models extracted by any other NAS work mentioned in our related works are not comparable with LTS since either the task metrics, e.g., translation quality (measured by bleu score) versus generative quality (measured by perplexity), or the underlying search objective (e.g., latency/memory minimization) is different. Please see answer number 3 to weakness 2 by reviewer KSpx for more details.
> >
> > 3.  **Weakness 3 (effect of training hyperparameters):** Please see our response number 3 to the comments by reviewer Zj8C for a detailed discussion on this topic and our new experiment.
> >
> > 4.  **Weakness 4 (adding [5] to related work):** We thank the reviewer for bringing this work to our attention. Perhaps the main differing factor between LTS and [5] is the target domain, i.e., authors of [5] focus on vision Transformers for image classification and propose a training-free proxy to rank architectures in terms of image classification accuracy. The proposed proxy is composed of two components, the Synaptic Diversity and the Synaptic Saliency, where the latter has the same formulation as the synflow proxy studied in our paper. The proxy in [5] can potentially be applicable to the language modeling domain due to its similarities to synflow which has a high correlation with perplexity (see Figure 5 in our paper). We will evaluate this proxy for the camera-ready version of the paper and append it to our list of low-cost proxies in Sections 3.1 and 4.2. Going in the other direction, it will be also interesting to study the use of non-embedding parameters as a proxy in the setup of [5].
> >
> > 5.  **Weakness 5 (open-source code):** Our code is already open-sourced and integrated as part of a large-scale library of NAS methods. However, we omitted the GitHub link for anonymity in the review period. Upon acceptance of the paper, we will provide the link in the paper and also open-source all pre-trained models. We believe the pre-trained models will be extremely useful for the community as a unified benchmark since there are no existing open-source NAS benchmarks for Transformers in language modeling (e.g., similar to NAS-bench for CNNs [I]).
> >
> > [A] [HAT: Hardware-aware transformers for efficient natural language processing](https://arxiv.org/pdf/2005.14187.pdf), ACL 2020.
> >
> > [B] [Primer: Searching for Efficient Transformers for Language Modeling](https://openreview.net/pdf?id=bzpkxS_JVsI), Neurips 2021.
> >
> > [C] [Zero-cost proxies for lightweight NAS](https://openreview.net/pdf?id=0cmMMy8J5q), ICLR 2021.
> >
> > [D] [Scaling laws for autoregressive generative modeling](https://arxiv.org/pdf/2010.14701.pdf), 2020.
> >
> > [E] [Scaling laws for neural language models](https://arxiv.org/pdf/2001.08361.pdf), 2020.
> >
> > [F] [Scaling laws for neural machine translation](https://openreview.net/pdf?id=hR_SMu8cxCV), ICLR 2022.
> >
> > [G] [Scaling Laws for Transfer](https://arxiv.org/pdf/2102.01293.pdf), 2022.
> >
> > [H][ Hyperband: A novel bandit-based approach to hyperparameter optimization](https://jmlr.org/papers/volume18/16-558/16-558.pdf), JMLR 2018.
> >
> > [I] [Nas-bench-201: Extending the scope of reproducible neural architecture search](https://arxiv.org/pdf/2001.00326.pdf), ICLR 2020.

---

> > > ### Comment · Reviewer_zV4x · 2022-08-07
> > > **Response**
> > >
> > > I'd like to thank the authors for testing some of my suggestions and answering my questions. Please incorporate the response to the paper where appropriate.
> > >
> > > Overall, the authors largely addressed my concerns, and I am increasing my score slightly. While I do share some of the concerns by other reviewers (such as the focus on autoregressive transformers only), I believe there are worthwhile contributions in this paper.

---

> > > > ### Author Response · Authors · 2022-08-07
> > > > **Thanks!**
> > > >
> > > > Thanks for the response! We are making sure to fully incorporate the response in the manuscript and the reviewer's comments have made the paper better.

---

### Official Review · Reviewer_Zj8C · 2022-07-09

**Rating:** 6
**Confidence:** 4
**Soundness:** 2 fair
**Presentation:** 4 excellent
**Contribution:** 3 good

**Summary:**

This paper tackles the problem of efficient deployment of autoregressive transformer models in resource constrained environment. Finding architectures that can solve the dual problem of latency and accuracy in such environments can be difficult. This problem gets exacerbated when one tries to deploy across multiple hardware platforms. NAS, via its SuperNet methodology,  has been proposed to manage this problem. However, SuperNet training is expensive and can lead to large memory requirement. Additionally, training supernets is non-trivial as subnets can interfere with each other and rank correlation between sub architectures is not preserved. Non SuperNet approach to NAS generally have an expensive architecture evaluation phase where accuracy on a training data is measured, which makes it difficult to use in practice.

This paper makes the following contribution in this problem domain -
1. The paper finds the surprising correlation that for autoregressive transformer model, the number of decoder parameters is a good proxy for validation perplexity. This allows for a very simple architecture evaluation phase.
2. Building on this insight, they develop a lightweight transformer search algorithm that combines evolutionary search with this simple accuracy proxy and real runtime measurement on hardware to develop a framework that provides a better accuracy-runtime pareto frontier than the traditional method of layer scaling.

**Questions:**

Questions -

Can the authors comment on Weakness (1), (2) and (3)?

**Ethics Review Area:**

["I don’t know"]

**Limitations:**

Have the authors adequately addressed the limitations and potential negative societal impact of their work? Yes

**Strengths And Weaknesses:**

Strengths -
1. Comparison with a large number of simple accuracy proxy methods from literature
2. Ablation studies like the one to understand the impact of topology of the model on accuracy and how it impacts the proposed proxy method for accuracy
3. The proposed solution can easily adapt to any hardware platform, while not losing its efficiency


Weaknesses -
1. The entire paper hinges on the idea that parameter count is a good proxy for validation perplexity. However, the validation of this claim is not done cleanly. The authors rely on training results on wikitext-103,  an extremely small dataset. Additionally, the use the same training hyperaparameter for different models to verify their claim for parameter count serving as a proxy for validation perplexity. Autoregressive models are trained on far larger datasets[2][3]. Infact, training on small datasets can have a significant impact on validation perplexity [2]. Additionally, dataset is expected to scale with size of the model.  Finally, [1] also shows that using the same training hyperparameters for models at different scale can lead to poor accuracy. These facts indicate that this claim is based on weak experimentation methodology. As a result, it becomes unclear if a more appropriate experiment method will lead to the same conclusion.

2. Additionally, using validation metric as a measure of quality of an autoregressive model is also questionable. Autoregressive models have shown immense benefit in generative, zero shot and few shot/prompt tuned tasks. Additionally, these are the tasks for which these models are being currently deployed in the industry (see cohere.ai, OpenAI, AI21 APIs for these models). Given that these tasks represent the way trained autoregressive models are consumed in the industry, measuring accuracy based on some of these tasks would make sense. Its unclear whether validation perplexity is a good proxy for these tasks or not. This further makes it difficult to understand the impact of this work and the usefulness of the result.

3. Their claims on impact of topology and relevance of the proxy metric for iso-parameter, different width/depth factor models is also suffering from similar methodology problem. As you make models deeper and thinner, initialization becomes extremely important to get good perplexity score. As a result, if the init of parameters is not explored, accuracy will suffer, making the claims harder to justify.

[1] https://www.microsoft.com/en-us/research/blog/%C2%B5transfer-a-technique-for-hyperparameter-tuning-of-enormous-neural-networks/

[2] https://arxiv.org/abs/2203.15556

[3] https://ai.googleblog.com/2020/02/exploring-transfer-learning-with-t5.html

---

> ### Author Response · Authors · 2022-08-02
> **Response to reviewer Zj8C (part 1)**
>
> We thank the reviewer for handling our manuscript and their insightful comments. Below we address all comments in detail. Please let us know if there are any further questions that we can help address during the interactive rebuttal period.
>
> 1.  **Weakness 1 (dataset size):** We note that our evaluations are not limited to WikiText-103. The main results of the paper, i.e., the high correlation of decoder parameter count with perplexity (Fig. 3) and Pareto-front extraction with NAS (Fig. 8) are also validated on the 1 billion word dataset  (LM1B) which is 10x larger than WikiText-103 and is among the largest publicly available language modeling datasets. While even larger datasets are available (e.g. The Pile) LM1B and WikiText103 (2x larger than WikiText2 studied in [1]) provide representative investigation while keeping experimentation cost reasonable. Specifically, since no previous NAS benchmark is available for autoregressive Transformers.
>
> 2.  **Weakness 1 (dataset should scale with model size [2,3]):** While this statement is generally true for large-scale models, our focus is on efficient Transformers. The range of model sizes in our search spaces is therefore far smaller than the ranges studied in [2,3]. Specifically, our randomly selected models in Figure 3 have between 8 to 70 million total parameters (2 to 45 million decoder parameters). Using the formula in [2], the maximum number of training tokens needed for us, i.e., for a model size of 70M, is 1.4B tokens, which is roughly the size of LM1B (1B).
>
> 3.  **Weakness 1 (training Hyperparameters):** The setup for our training hyperparameters is based on the evidence provided in highly cited works as follows:
> 	-   **Learning rate (LR):** [C] states the following observation in Appendix D.6: “*We find that, as long as the learning rate is not too small and does not decay too quickly, performance does not depend strongly on learning rate*”. They also find that “*larger models require a smaller learning rate to prevent divergence, while smaller models can tolerate a larger learning rate*“, and provide a rule-of-thumb for setting the optimal LR in Equation D.1. Adopting this rule, the optimal LRs changes negligibly for our range of model sizes, e.g., 2-45M decoder parameters in Fig. 3. This suggests that the same training setup can be used for all architectures in the search-space.
> 	    Nevertheless, our curiosity was piqued by the reviewer’s comment and we conducted a new experiment. We sweep LR over [0.0001, 0.001, 0.01, 0.1] for 90 models with 2-16 layers and 2-65M parameters. We pick the optimal LR for each model which results in the lowest perplexity. We remeasure the correlation between new perplexities and decoder parameter county proxy. Our experiment shows two important points: 1) the optimal LR is equal to what we used in our original experiments for the vast majority of the architectures, and 2) the ranking of architectures after convergence remains largely unchanged, leading to a correlation of 0.93 with decoder parameter count (compared to 0.96 when using the same LR for all models).
>
> 	-   **LR scheduler:** The scaling laws paper  [C] also shows that the choice of LR scheduler does not have a significant effect on final model performance (see Appendix D.6 of [C]). GPT3 [B] also adopts the same LR scheduler for all models, regardless of their size, which further validates that changing the scheduler will not affect final performance.
>
> 	-   **Batch size:** For the range of model sizes studied in this paper, prior work adopts the same batch size (see Table 2.1 in GPT3 [B]), which suggests there is no significant benefit in optimizing the batch size per architecture.
>
> 	-   We would like to emphasize that while hyperparameter optimization (HPO) is an important and active area of research, due to the extremely high cost of NAS + HPO, the vast majority of papers in NAS use the same hyperparameters for all models [D,E,F,G,H]. Many impactful papers in NAS rely on supernet training, which inherently uses the same training setup for all sub-architectures. Additionally, upcoming papers [L] that study the scaling phenomenon in Transformers use the same training setup for all models. Finally, many existing benchmarks for NAS [I,J,K] which are widely used in the community also adopt the same training hyperparameters for all models due to the extremely high cost of HPO.
>
> 	-   **Bottomline:** The reviewer’s point is well-taken but given the expensive cost of joint NAS+HPO, it is difficult to study and it remains an open problem for the AutoML community. We hope we have convincingly shown that HPO is *not crucial* for our particular search space and task pairing and will not affect our empirical findings. We have added the above new experiment on learning rate optimization along with the discussions on HPO to the revised manuscript, Appendix B. We would love to engage with the reviewer more on this and know their further thoughts.

---

> > ### Author Response · Authors · 2022-08-02
> > **Response to reviewer Zj8C (part 2)**
> >
> > 4.  **Weakness 2 (other metrics):** The following sentence from GPT3 paper [B] suggests there is a strong correlation between perplexity (loss) and performance on other tasks *"...improvements in cross-entropy loss lead to consistent performance gains across a broad spectrum of natural language tasks.”*. Nevertheless, it is worth noting that small autoregressive Transformers, i.e., those studied in this paper, are “single-task” models trained and used specifically for “generative (language) modeling”. For this purpose, perplexity is the best metric as it directly correlates with the training cross-entropy loss (see for example that perplexity is used as the metric in the original Transformer-XL paper [M]). The great performance of autoregressive transformers in “multi-task” settings, i.e., zero-shot and few-shot tuned tasks are applicable to Billion-parameter scale models. As a concrete example, the good performance of GPT2/GPT3 on few-shot/zero-shot tasks is only true for >1 Billion parameter model as shown in Figure 1 in [A] and Figures 3.2, 3.3., and 3.4 in [B]. These models are not the focus of this paper as we aim to find small models that can run efficiently on commodity hardware.
> >
> > 5.  **Weakness 3 (initialization):** Please note that our initialization is not the same for different architectures in Figure 7. Specifically, we adopt the proposed initialization in the GPT2 paper [A] which scales the initialized weights of residual layers by 1/sqrt(n) where n is the number of residual layers (depth of the model). This initialization scheme is also used in the GPT-3 paper [B] and has shown success in stabilizing training for a wide range of models up to 100 layers by taking into account the topology. While exploring other variants of initialization is certainly interesting, it requires thousands of GPU hours for the scale of experiments conducted in this paper (>2800 trained models). We, therefore, chose to adopt the initialization method proposed in the prior art which provides an agreeable degree of customization for the underlying topology.
> >
> > [A] [Language models are unsupervised multitask learners](https://d4mucfpksywv.cloudfront.net/better-language-models/language_models_are_unsupervised_multitask_learners.pdf), OpenAI blog 2019.
> >
> > [B] [Language models are few-shot learners](https://arxiv.org/pdf/2005.14165.pdf), Neurips 2020.
> >
> > [C] [Scaling laws for neural language models](https://arxiv.org/pdf/2001.08361.pdf), 2020.
> >
> > [D] [HAT: Hardware-aware transformers for efficient natural language processing](https://arxiv.org/pdf/2005.14187.pdf), ACL 2020.
> >
> > [E] [Primer: Searching for Efficient Transformers for Language Modeling](https://openreview.net/pdf?id=bzpkxS_JVsI), Neurips 2021.
> >
> > [F] [Training-free Transformer Architecture Search](https://arxiv.org/pdf/2203.12217.pdf), CVPR 2022.
> >
> > [G] [Autoformer: Searching transformers for visual recognition](https://openaccess.thecvf.com/content/ICCV2021/papers/Chen_AutoFormer_Searching_Transformers_for_Visual_Recognition_ICCV_2021_paper.pdf), ECCV 2021.
> >
> > [H] [Exploring the Limits of Transfer Learning with a Unified Text-to-Text Transformer](https://arxiv.org/pdf/1910.10683.pdf), JMLR 2020.
> >
> > [I] [Nas-bench-201: Extending the scope of reproducible neural architecture search](https://arxiv.org/pdf/2001.00326.pdf), ICLR 2020.
> >
> > [J] [Nas-bench-301 and the case for surrogate benchmarks for neural architecture search](https://openreview.net/pdf?id=1flmvXGGJaa), 2021.
> >
> > [K] [Surrogate NAS benchmarks: Going beyond the limited search spaces of tabular NAS benchmarks](https://openreview.net/pdf?id=OnpFa95RVqs), ICLR 2022.
> >
> > [L] [Scaling laws for neural machine translation](https://openreview.net/pdf?id=hR_SMu8cxCV), ICLR 2022.
> >
> > [M] [Transformer-xl: Attentive language models beyond a fixed-length context](https://arxiv.org/pdf/1901.02860.pdf), ACL 2019.

---

> > > ### Comment · Reviewer_Zj8C · 2022-08-08
> > > **Thank you for the detailed responses, updated the evaluation score**
> > >
> > > Thanks for your careful rebuttal and revision, which I highly appreciate. After reading the rebuttal and revision, I decide to raise the score of my evaluation.

---

> > > > ### Author Response · Authors · 2022-08-08
> > > > **Thank you for the note**
> > > >
> > > > Thank you so much for your kind note and we are glad we were able to address your concerns.

---

### Official Review · Reviewer_rzmB · 2022-07-13

**Rating:** 5
**Confidence:** 3
**Soundness:** 3 good
**Presentation:** 3 good
**Contribution:** 3 good

**Summary:**

The paper studies neural architecture search over autoregressive transformer models, and proposes a training-free approach to search the Pareto frontier of multiple objectives: latency, memory and perplexity. The proposed approach is based on an important finding that the perplexity is highly correlated with the number of parameters in the decoder, which offers a cheap and convenient way of model evaluation.


**Questions:**

Please address the comments above.

**Ethics Review Area:**

["I don’t know"]

**Limitations:**

yes

**Strengths And Weaknesses:**

Strengths:
- The paper proposes a training-free NAS approach to discover the Pareto front of auto-regressive transformer architectures, which can significantly reduce the searching cost in designing efficient models.
- The authors provide numerous empirical results to justify the training-free proxy, together with its relation with the latency and memory of target devices.
- Well written and easy to follow.

Weakness:
There are still several concerns with the finding that the perplexity is highly correlated with the number of decoder parameters.
- According to Figure 4, the correlation decreases as top-10% architectures are chosen instead of top-100%, which indicates that the training-free proxy is less accurate for parameter-heavy decoders.
- The range of sampled architectures should also affect the correlation. For instance, once the sampled architectures are of similar sizes, it could be more challenging to differentiate their perplexity and thus the correlation can be lower.


Detailed Comments:
1. Some questions regarding Figure 4: 1) there is a drop of correlation after a short period of training, which goes up with more training iterations; 2) the title "Top-x%" should be further explained;
2. Though the proposed approach yields the Pareto frontier of perplexity, latency and memory, is there any systematic way to choose a single architecture given the target perplexity?

---

> ### Author Response · Authors · 2022-08-02
> **Response to reviewer rzmB**
>
> We thank the reviewer for handling our manuscript and their kind words regarding the strengths and contributions of our work. Below we address the weaknesses and comments in detail. Please let us know if there are any further questions that we can help address during the interactive rebuttal period.
>
> 1.  **Weakness 1 (rank correlation over top-10% in Figure 4):** It is true that the rank correlation over top-10% is relatively lower compared to top-100%. However, we would like to emphasize that our proxy archives a spearman rank correlation of **0.68** over top-10% which is significantly higher than the training-free proxies in other work, e.g., the best proxy in [A] achieves a rank correlation of **0.42** or **0.14** over top-10% depending on the benchmark (see [A] Section 5.3). Nevertheless, [A] one of the early pioneering work which brought a lot of attention to such proxies, shows that even much smaller top-10% rank correlations can lead to a successful NAS.
> Crucially, the common ratio for our proxy over top-10% is **78%** which shows that the proxy is able to identify top-performing candidate models with high accuracy. Just a reminder that common ratio over top-10% means the set intersection of the top-10% ground truth set and the set suggested by our proxy to be the top-10%. We apologize for a previous error in the topk titles of Figure 4 which had a mismatch with the reported rank correlation values. We have corrected the Figure in the revised paper.
>
> 2.  **Weakness 2 (effect of sampled range on correlation):** Figure 10 in Appendix C captures this effect by studying the ranking correlation of our proxy with ground-truth perplexity when evaluated on architectures with very small differences in their parameter count (<5M). As seen, the rank correlation decreases compared to evaluations on a wider range of architectures. However, the lowest correlation value (0.71) is still significantly larger compared to the correlations obtained by other zero-cost proxies, e.g., in [A] and [B].
>
> 3.  **Comment 1.1 (intuition for the drop of correlation in partial training (Figure 4)):** As shown in Figure 5, low-cost proxies which evaluate the model at initialization have a high correlation with the ground-truth ranking of architectures. The high correlation after very free training iterations (tau = 1.25%) is perhaps a by-product of the same phenomenon. However, as training progresses, the fluctuations in the batch loss (particularly in the rapid warmup phase of the learning rate), can lead to inaccurate estimates of the learning curve, which results in unreliable ranking of models based on partially observed training. After training stabilizes (tau > 10%), the partial training correlation steadily increases.
>
> 4.  **Comment 1.2 (defining top-k):** We divide architectures into topk bins based on their perplexity after full training, i.e., topk corresponds to the models which achieve the topk% of perplexities. We have added an explanation for top-k in the caption of Figure 4 as well as in the definition of the performance criteria in Section 4.1 of the revised paper.
>
> 5.  **Comment 2 (how to choose a single architecture):** In the context of multi-objective NAS, the pareto frontier model that satisfies a target perplexity is a model where any modification to its topology would either result in a worse perplexity, and/or latency and memory. Therefore, if the user has a target perplexity of 30, then in Figure 8 one simply draws a horizontal line from ppl=30 on the y-axis and picks the nearest model below this line on the Pareto curve. By definition of a  [pareto-frontier](https://en.wikipedia.org/wiki/Pareto_front#:~:text=In%20multi%2Dobjective%20optimization%2C%20the,is%20widely%20used%20in%20engineering), that will be the best model in terms of latency and memory which achieves the target (or a slightly better) ppl.
>
> [A] [Zero-cost proxies for lightweight NAS](https://openreview.net/pdf?id=0cmMMy8J5q), ICLR 2021.
>
> [B] [A Deeper Look at Zero-Cost Proxies for NAS](https://iclr-blog-track.github.io/2022/03/25/zero-cost-proxies/), ICLR blog 2022.

---

> ### Comment · Reviewer_rzmB · 2022-08-08
> **response**
>
> I thank the authors for providing the detailed feedback for my concerns, though some of them are more like better clarified rather than addressed. After reading the rebuttal, I decide to keep my original rating unchanged.

---

> > ### Author Response · Authors · 2022-08-08
> > **Thank you for the note**
> >
> > Thank you so much for reading our response and we are happy we were able to clarify any ambiguities.

---

### Author Response · Authors · 2022-08-02
**Brief overview of author response**

We thank the reviewers for their valuable comments and suggestions. Specifically, we are thankful to reviewers for acknowledging the significance of our training-free efficient search and the potential impact on the field. We are also glad that the reviewers found our numerous experiments, ablation studies, and analyses to be thorough and well-written. To address any remaining concerns or questions, below we provide detailed answers to each reviewer. In summary, we address the following key points in our drafted response:
-   Clarification of training routine and datasets
-   Clarification of the rationale behind the evaluation metric and baselines
-   New experiments that show the potential application of our training-free proxies to other domains, e.g., Neural Machine Translation (NMT).

We have also revised our manuscript to integrate the comments by the reviewers and our responses. All new/edited text is marked in the color Blue.

---

### Author Response · Authors · 2022-08-08
**Following up on the rebuttal responses**

Dear reviewers,

Thank you so much for handling our manuscript and providing insightful comments that helped improve our work. Given the remaining short time for the discussion period, we would really appreciate your opinion on our response and it will be extremely valuable to us to know your feedback. We are looking forward to hearing from you and thanks once again for your time and effort.

---

### Meta-Review · Area_Chair_tLo6 · 2022-08-23

**Recommendation:** Accept
**Confidence:** Less certain

**Metareview:**

Building on the novel observation that there is a strong correlation between model quality and number of parameters in the decoder of autoregressive Transformers, this work proposes a training-free NAS algorithm for this class of models.

The main concerns raised by reviewers are related to the key observation the paper is built on (this correlation), that may not necessarily hold in all situations. However, authors were able to clarify important points, provide additional motivations and empirical results, eventually leading to all reviewers leaning towards acceptance.

Considering the popularity of this class of models, I believe this work should be of significant interest to researchers and practitioners in the field, and I am thus recommending acceptance.

**Award:**

No

---

### Decision · Program_Chairs · 2022-09-14

Accept